# ReLayout: Integrating Relation Reasoning for Content-aware Layout Generation with Multi-modal Large Language Models

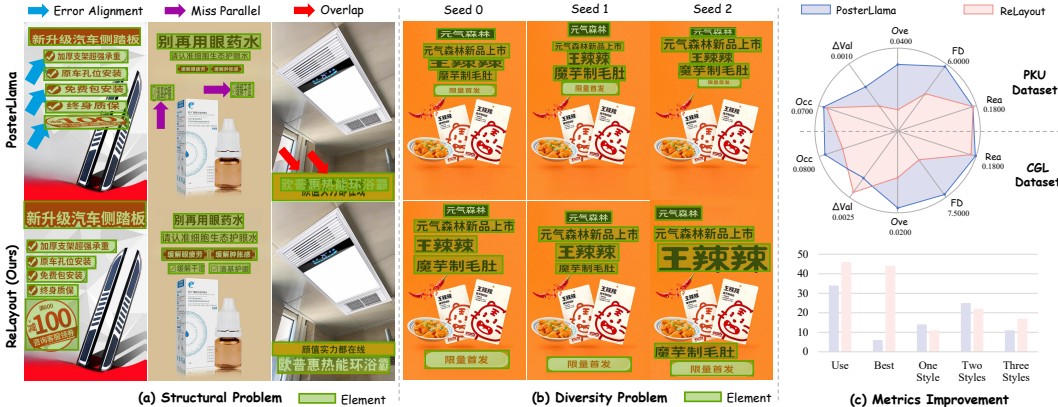

Figure 1: Comparison with current SOTA PosterLlama (Seol et al., 2024). (a) PosterLlama exhibits structural issues: error alignment, missing parallelism, and overlap. (b) Our method generates more diverse layouts across different seeds. (c) In the top figure, **lower values** indicate better performance across all metrics. In the bottom figure, all metrics are from user studies, including the number of usable images, best images, and images with one, two, or three styles. Evaluations are based on 50 images from PKU dataset using seeds 0, 1, and 2, with the first two metrics taken from seed 0.

## ABSTRACT

Content-aware layout aims to arrange design elements appropriately on a given canvas to convey information effectively. Recently, the trend for this task has been to leverage large language models (LLMs) to generate layouts automatically, achieving remarkable performance. However, existing LLM-based methods fail to adequately interpret spatial relationships among visual themes and design elements, leading to structural and diverse problems in layout generation. To address this issue, we introduce ReLayout, a novel method that leverages relation-CoT to generate more reasonable and aesthetically coherent layouts by fundamentally originating from design concepts. Specifically, we enhance layout annotations by introducing explicit relation definitions, such as region, saliency, and margin between elements, with the goal of decomposing the layout into smaller, structured, and recursive layouts, thereby enabling the generation of more structured layouts. Furthermore, based on these defined relationships, we introduce a layout prototype rebalance sampler, which defines layout prototype features across three dimensions and quantifies distinct layout styles. This sampler addresses uniformity issues in generation that arise from data bias in the prototype distribution balance process. Extensive experimental results verify that ReLayout outperforms baselines and can generate structural and diverse layouts that are more aligned with human aesthetics and more explainable.

# 1 INTRODUCTION

Layout is an essential part of graphic design, aiming to convey information through the appropriate arrangement of elements such as logos and texts. Due to its importance, layout has various applications, spanning scenarios like documents (Li et al., 2019a), UIs (Raneburger et al., 2012; Deka et al., 2017), magazines (Yang et al., 2016; Tabata et al., 2019) and posters (Guo et al., 2021; Lin et al., 2023). Among these, when the main visual element flows into an application, such as advertising posters, achieving harmony between the arrangement of elements and the canvas becomes one of the key goals. We call layout generation under the above condition content-aware layout generation.

This field is particularly challenging because it requires the integration of design elements, such as logos and text, with visual content to produce layouts that are both usable and aesthetically pleasing. Furthermore, the model needs to generate diverse layouts to ensure diversity. To address these challenges, researchers have proposed various methods (Zheng et al., 2019a; Horita et al., 2024; Hsu et al., 2023; Zhou et al., 2022) based on generative models (Goodfellow et al., 2020; Kingma, 2013; Ho et al., 2020) to enhance the quality of generated layouts. Among these methods, RALF (Horita et al., 2024), as a transformer-based (Vaswani, 2017) method, has achieved notable advancements. It adopts a retrieval augmentation method to mitigate the data scarcity problem. Nevertheless, it treats layout generation only as a numerical problem, failing to capture the semantics, which prevents the model from generating visually and textually coherent layouts.

Recently, three LLM-based methods (Lin et al., 2024; Seol et al., 2024; Hsu & Peng, 2025) have emerged, aiming to leverage the ability of large language models to generate high-quality layouts. For instance, LayoutPrompter (Lin et al., 2024) employs dynamic exemplar selection to generate layouts without requiring training but cannot take a canvas image as input, thereby missing out on a significant amount of information. PosterLlama (Seol et al., 2024), as the current SOTA, trains a mutli-modal large language model (MLLM) to generate visually and textually coherent layouts. PosterO (Hsu & Peng, 2025) introduces a training-free method that leverages retrieval augmentation and design intent maps to generate better layouts. However, these methods are limited to predicting element-level coordinates (e.g., 'where to place' each individual element). They lack a structural-level understanding that explicitly models the logical relationships between elements, such as recursive grouping and hierarchy. This limitation hinders the incorporation of high-level design concepts, maintaining overall visual harmony and ensuring layout diversity, thereby leading to two critical issues in layout generation: (1) structural problem, in which related elements fail to maintain proper spatial relationships, as illustrated in Figure 1(a), where PosterLlama produces overlapping elements, incorrect alignments, and fails to capture parallel relationships; and (2) diversity problem, where the generated layouts lack the rich structural variation found, as shown in Figure 1(b), where these methods, without explicit modeling of element relationships, degrade to similar structural arrangements. Also, the top part of Figure 1(c) shows that our method generates better layouts, and the bottom part demonstrates its ability to produce more diverse styles for the same canvas under different seeds.

To address this issue, we introduce ReLayout. Inspired by CoT, we internalize sequential reasoning directly into the supervision data rather than relying on inference prompts, bridging the gap between static layout annotations and the dynamic design process. It decomposes the raw dataset into a structured reasoning path: first identifying salient boundaries, then defining regional structures, and finally placing elements. This transforms the layout generation task from a mere numerical regression problem into a step-by-step logical derivation. Consequently, MLLMs learn to mimic this expert logic, autonomously generating intermediate relation tokens during inference for more coherent layouts. Additionally, we introduce the layout prototype rebalance sampler, which quantifies the layout prototype into a three-dimensional feature space of saliency, region, and margin between elements based on the layout relation-CoT construction. By integrating feature clustering with weighted sampling, the sampler enables balanced learning of diverse layout prototypes, thereby enhancing the diversity of generated layouts. User studies and visualization demonstrate that ReLayout outperforms state-of-the-art methods, achieving significant improvements in usability and diversity. In summary, our contributions are as follows:

- We propose ReLayout, a relation-CoT paradigm designed to address hierarchical layout design challenges, specifically tackling structural and diversity problems via explicit spatial relations and layout prototype balancing.

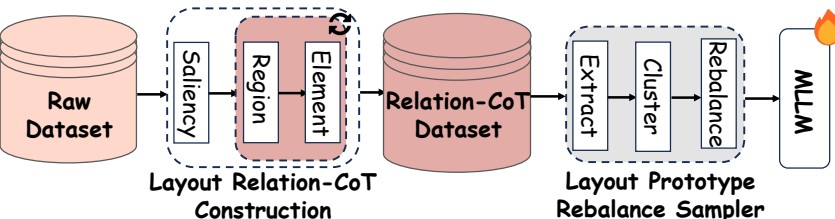

Figure 2: Pipeline of ReLayout. We adopt the layout relation-CoT construction to add relation annotations on raw datasets. Then we use the layout prototype rebalance sampler to adjust the distribution of the new dataset for training.

- We introduce a layout relation-CoT construction mechanism that decomposes layout element relationships into a hierarchical structure while incorporating element relation annotations into existing layout datasets, which we will release with richer layout information.
- We develop a layout prototype rebalance sampler, which quantifies layout prototypes through feature clustering and employs weighted sampling to ensure adaptability across diverse real-world scenarios.

## 2 RELATED WORK

### 2.1 AUTOMATIC LAYOUT GENERATION

**Content-agnostic:** Content-agnostic layout generation focuses on layouts without relying on specific content. LayoutGAN (Li et al., 2019b) is the first method to introduce GAN for this task; in addition, approaches involving VAE (Jiang et al., 2022; Jyothi et al., 2019) or Diffusion models (Chai et al., 2023; Inoue et al., 2023) have also been employed to solve this task. LayoutNUWA (Tang et al., 2024), an LLM-based method using HTML format, also shows strong performance.

**Content-aware:** Content-aware layout generation emphasizes not only layout quality, like content-agnostic methods, but also its harmony with the canvas. (O'Donovan et al., 2014) evaluated layout quality using a function that explicitly incorporates various design principles and constraints. (Zheng et al., 2019b) considered the empty space (or negative space) and overlapping among design elements. ContentGAN (Zheng et al., 2019a) first addressed the above problem. Later works, starting from CGL-GAN (Zhou et al., 2022), commonly use saliency maps. DS-GAN (Hsu et al., 2023) uses a CNN-LSTM model to balance graphic and content-aware metrics. RADM (Li et al., 2023a) is the first diffusion-based method to incorporate textual content into layout tasks. RALF (Horita et al., 2024) leverages a retrieval augmentation method to mitigate the data scarcity problem. Powered by LLMs, LayoutPrompter (Lin et al., 2024), PosterO (Hsu & Peng, 2025), and PosterLlama (Seol et al., 2024) demonstrate remarkable capabilities in the field of layout generation. LayoutPrompter uses prompt selection for training-free generation, PosterO introduces intent maps to avoid salient objects, and PosterLlama fine-tunes the model for coherent visual-textual layouts.

Unlike previous LLM-based methods, our method explicitly represents relationships and decomposes a layout into smaller, structured, and recursive layouts. This leads to layouts that is both more visually appealing and explainable.

### 2.2 MULTI-MODAL LARGE LANGUAGE MODELS

**Advancements:** LLMs have demonstrated remarkable capabilities in natural language understanding. Based on this, MLLMs have achieved remarkable progress by integrating cross-modal data including visual, auditory, and so on (Li et al., 2023b; Radford et al., 2021), thereby significantly expanding their range of applications, such as GPT-4 (Achiam et al., 2023) and Gemini (Team et al., 2023), as well as open-source models like InternVL (Chen et al., 2024) and QwenVL (Team, 2025).

**Techniques:** In recent years, several techniques have enhanced LLM capabilities. Few-shot (Brown et al., 2020) allows models to adapt to new tasks with few examples. CoT (Wei et al., 2022) improves reasoning by guiding models to break down complex problems step by step. LoRA fine-tuning (Hu

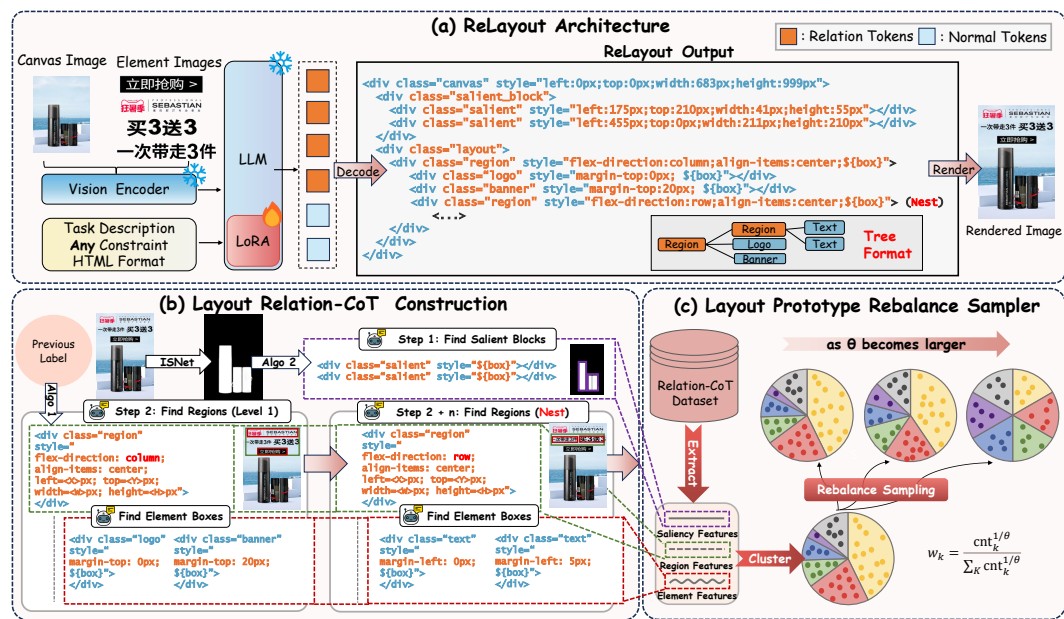

Figure 3: (a) is ReLayout training process and its output distinction from previous methods. The bottom part is two key components of ReLayout. (b) illustrates the relation labels construction logic. (c) represents the layout dataset resampling process, which adjusts the dataset distribution to achieve a more balanced layout dataset.

et al., 2022) efficiently adapts models by adding small trainable matrices, reducing memory and computation costs while maintaining strong performance.

In this work, we adapt several MLLMs for layout generation and enhance output formats inspired by CoT to produce more reasonable layouts. Results validate the effectiveness of ReLayout.

## 3 METHODS

### 3.1 OVERVIEW

Given a set of constraints, our goal is to generate a well-arranged layout. A layout $\mathcal{L}$ can be represented as a set of $N$ elements: $\mathcal{L} = \{\mathbf{e}_1, \ldots, \mathbf{e}_N\} = \{(c_1, \mathbf{b}_1), \ldots, (c_N, \mathbf{b}_N)\}$, where each element $e_i$ consists of its class $c_i$ and corresponding bounding box $\mathbf{b}_i = [x_i, y_i, w_i, h_i]$. In our work, multimodal inputs are a canvas image $\mathbf{C}$ and foreground elements $\mathcal{F} = \{(\mathbf{t}_i, \mathbf{p}_i)\}_{i=1}^N$, where $\mathbf{t}_i$ represents text and $\mathbf{p}_i$ represents an element image.

The pipeline of ReLayout, shown in Figure 2, consists of two key components: layout relation-CoT construction and layout prototype rebalance sampler. The layout relation-CoT construction explicitly models the layout relations from three aspects: margin between elements, region, and saliency. These relations will be used for the training of the MLLM to enhance the model's usability. Furthermore, these explicit relation models enable us to balance the samples in different clusters from the perspective of design styles, so as to achieve better optimization and diverse results. The inference procedure is illustrated in Figure 3(a). Unlike previous layout generation methods based on LLMs to directly generate layout coordinates, our method first predicts the structured relations (highlighted in orange) and then generates the layout coordinates based on the provided canvas image $\mathbf{C}$ and foreground elements $\mathcal{F}$.

### 3.2 LAYOUT RELATION-COT CONSTRUCTION

To fully leverage the extensive knowledge of LLMs in layout design, we choose HTML to represent layouts. However, unlike previous LLM-based methods that represent layouts using HTML (Lin

et al., 2024; Seol et al., 2024), we introduce two types of relation spaces: **region** (including margin between elements) and **saliency** (see Figure 3(b)). These relational spaces are designed to address the shortcomings of previous methods, which often generate layouts that are poorly structured and lack human aesthetic appeal.

**Region:** Caused by the fact that LLMs are inherently more sensitive to highly structured data, we introduce region. Region $\mathcal{R}$ serves as the fundamental unit of spatial arrangement, with its internal structure adhering to a single direction pattern. It can be understood as individual small layouts, similar to the structure of a tree. Thus it is both **nestable** and **recursive**. This makes the layout annotations formed by it highly structured, allowing the generation of complex overall arrangements through simple construction rules.

Region is defined by three key properties: $\mathcal{R} = (d, a, \mathbf{b})$, where $d$ is the flex-direction, representing the arrangement direction of elements within the region: $d \in \{row, column\}$, $a$ represents align-items, and $\mathbf{b}$ represents the region's position and size. As illustrated in Step 2 and Step $2 + n$ of Figure 3(b), regions are constructed step by step. We use Figure 4 as examples to describe the specific steps of constructing our region. (1) We first perform the x-axis and y-axis projection operations on each element. (2) Then we analyze the IoD (Intersection over Detection) (Yu et al., 2020) matrix of projections to group bounding boxes into $G_x$ (x-axis groups) and $G_y$ (y-axis groups), where IoD is defined as the intersection between the detection

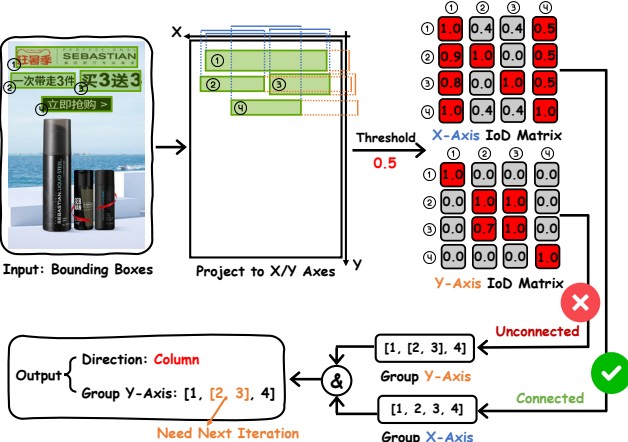

Figure 4: Visualization of region direction estimation heuristic algorithm in the layout relation-CoT construction.

box and the ignored region divided by the area of the detection box. (3) Based on the group counts, we determine the layout direction. At this point, we have obtained the direction of the level-1 region in Step 2 of Figure 3(b). Finally, we only need to recursively apply this process to each group to further subdivide the region, constructing a hierarchical structure like Step 2 + n of Figure 3(b). Detailed steps of the heuristic algorithm are given in the supplementary material.

Furthermore, parallel $\mathcal{P}$ (see the second column of Figure 1(a)) is a specialized type of region, sharing the same fundamental attributes. It is typically employed for the parallel presentation of two or more related elements. These elements maintain uniform visual sizes and align along a designated axis (either row or column) to ensure consistency and symmetry within the layout.

For each element within a region, we introduce an additional attribute, *margin*, to represent relative position, i.e., the spacing between elements. When the region is arranged in a row, this attribute is defined as *margin-left*, whereas in a column, it is specified as *margin-top*. Using this property, we can effectively control the overall layout compactness.

**Saliency:** Inspired by the goal that designers usually avoid placing elements over salient objects, we introduce salient blocks $\mathcal{S}$ to help the model better grasp features of salient objects. These blocks are represented as a series of bounding boxes and are integrated into an HTML-based representation. To detect these salient blocks, we propose an algorithm that efficiently identifies salient areas through integral image computation. This algorithm, detailed in the supplementary material, progressively selects non-overlapping rectangular regions by evaluating their saliency scores based on the density of white and black pixels, ensuring the captured regions align with natural visual attention. The following ablation studies also explains that adding salient blocks is crucial.

**Sequence formalization:** Our input sequence comprises a primary instruction, a task description (e.g., `"layout generation with given class"`), and an input HTML format. Four mask tokens (`<X>`, `<Y>`, `<W>`, `<H>`) are introduced to facilitate their prediction.

We combine the Saliency and Region components into a unified HTML format as the output sequence (see Table 8 in the supplementary material).

### 3.3 LAYOUT PROTOTYPE REBALANCE SAMPLER

Building upon layout relation-CoT annotations, we propose the layout prototype rebalance sampler to address the issue of limited diversity in previous methods. By the process, our method ensures a more even distribution across diverse layout prototypes, providing the model with greater opportunities to learn and generalize over a broader range of layouts. As shown in Figure 3(c), our layout prototype rebalance sampler consists of three key operations: feature extraction, feature clustering, and rebalance sampling. Below, we provide a detailed explanation of each operation.

**Feature extraction:** The $i^{\text{th}}$ layout prototype is to be primarily characterized by three dimensions: $\{\mathcal{S}_i, \mathcal{R}_i, \mathcal{E}_i\}$. The set of saliency bounding boxes in the $i^{\text{th}}$ layout is denoted as $\mathcal{S}_i$, given by: $\mathcal{S}_i = \left\{\mathbf{b}_{i,j}^{\text{s}}\right\}_{j=1}^{r_i}$. The number of saliency bounding boxes in layout $L_i$ is given by $r_i \in \{1, 2, 3, 4\}$. The saliency feature vector $\mathbf{f}_i^{\text{s}}$ for layout $L_i$ captures the weighted center of all saliency boxes in Equation 1. We define the set of regions in a layout as $\mathcal{R}_i = \{\mathbf{b}_{i,j}^{\text{r}}, d_{i,j}\}_{j=1}^{s_i}$, where $d_{i,j} \in \{\text{row}, \text{column}\}$ represents the region's alignment direction. Then, we extract statistical features $\mathbf{f}_i^{\text{r}}$ from $\mathcal{R}_i$ to describe their spatial distribution in Equation 2.

$$\mathbf{f}_i^{\text{s}} = \begin{pmatrix} \dfrac{\sum_{j=1}^{r_i} \left(x_{i,j}^{\text{s}} + \frac{w_{i,j}^{\text{s}}}{2}\right) w_{i,j}^{\text{s}} h_{i,j}^{\text{s}}}{\sum_{j=1}^{r_i} w_{i,j}^{\text{s}} h_{i,j}^{\text{s}}} \\[2em] \dfrac{\sum_{j=1}^{r_i} \left(y_{i,j}^{\text{s}} + \frac{h_{i,j}^{\text{s}}}{2}\right) w_{i,j}^{\text{s}} h_{i,j}^{\text{s}}}{\sum_{j=1}^{r_i} w_{i,j}^{\text{s}} h_{i,j}^{\text{s}}} \end{pmatrix} \in \mathbb{R}^2. \quad (1) \qquad \mathbf{f}_i^{\text{r}} = \begin{cases} s_i = |\mathcal{R}_i|, \\[1em] \sigma_i^{\text{x}} = \text{std}\left(\left\{x_{i,j}^{\text{r}} + \frac{w_{i,j}^{\text{r}}}{2}\right\}_{j=1}^{s_i}\right), \\[1em] \sigma_i^{\text{y}} = \text{std}\left(\left\{y_{i,j}^{\text{r}} + \frac{h_{i,j}^{\text{r}}}{2}\right\}_{j=1}^{s_i}\right), \\[1em] n_i^{\text{row}} = \sum_{j=1}^{s_i} \mathbb{I}\left(d_{i,j} = \text{row}\right), \\[1em] n_i^{\text{column}} = \sum_{j=1}^{s_i} \mathbb{I}\left(d_{i,j} = \text{column}\right), \end{cases} \quad (2)$$

We define the element set of the $i^{\text{th}}$ layout as $\mathcal{E}_i = \{c_{i,j}\}_{j=1}^{t_i}$, where $t_i$ is the total number of

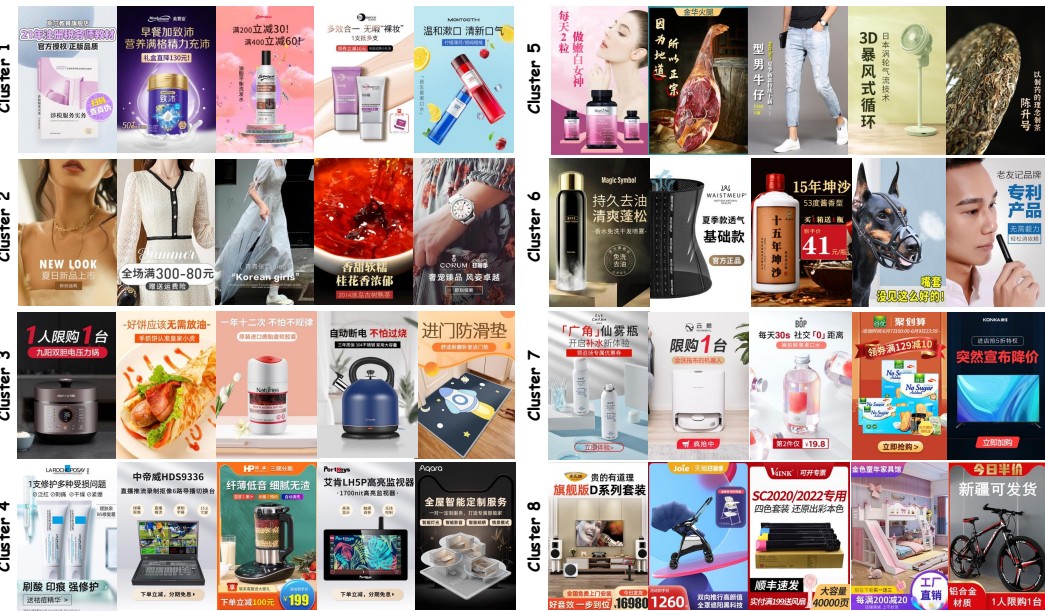

Figure 5: Example layouts from three clusters showing distinct characteristics in saliency, region, and element dimensions.

elements, and $c_{i,j}$ represents the category of the $j^{\text{th}}$ element in the $i^{\text{th}}$ layout. We believe that the layout is highly related to the types and numbers of elements. Therefore, we define element-level features as follows: $\mathbf{f}_i^{\text{e}} = \left(\sum_{j=1}^{t_i} \mathbb{I}(c_{i,j} = c_k)\right)_{k=1}^{K}$, where $K$ denotes the predefined number of element categories (e.g., text, logo) in the dataset.

**Feature cluster:** The final feature is constructed by weighted concatenation of the three feature dimensions:

$$\mathbf{f}_i = \alpha \mathbf{f}_i^{\text{s}} \oplus \beta \mathbf{f}_i^{\text{r}} \oplus \gamma \mathbf{f}_i^{\text{e}} \tag{3}$$

Using these aggregated feature vectors, we apply K-means clustering with 8 clusters to group similar layouts for analysis. Figure 5 illustrates eight layout clusters. Cluster 1 is characterized by a centered salient object flanked by two regions, one of which is typically in a column arrangement. Cluster 2 features a salient object that dominates the canvas, generally supported by one column region and top/bottom banners. Cluster 3 also centers the salient object but contains only a single region, usually encompassing two elements. Cluster 4 is defined by a complex, multi-element structure. Cluster 5 highlights layouts where elements are predominantly horizontally aligned. Cluster 6 contains layouts where the salient object is prominently situated on the left side of the canvas. Cluster 7's regions are uniquely distributed above and below the salient object. Cluster 8 is notable for incorporating numerous banner elements that adhere tightly to the edges of the canvas.

**Rebalance sampling:** After obtaining 8 clusters, we apply weighted sampling to balance their influence and avoid dominance by larger ones. Specifically, we assign a sampling weight to each cluster based on its size:

$$\mathbf{w} = \frac{\mathbf{cnt}^{1/\theta}}{\|\mathbf{cnt}^{1/\theta}\|_1}, \tag{4}$$

where $\|\mathbf{cnt}^{1/\theta}\|_1 = \sum_{k=1}^{K} \mathbf{cnt}_k^{1/\theta}$ and $\mathbf{cnt}_k$ represents the number of layouts in cluster $k$, and $\theta$ is a hyperparameter that controls the distribution of weights. Larger $\theta$ makes sampling more uniform, helping rare clusters be seen more, but too large may distort the data. Smaller $\theta$ favors large clusters and keeps the original distribution, but may miss rare cases.

## 4 EXPERIMENTS

### 4.1 DATASETS

We use two publicly available e-commerce datasets, CGL (Zhou et al., 2022) and PKU (Hsu et al., 2023). The PKU dataset includes three element categories: Logo, Banner, and Text, while the CGL dataset has an additional category called Embellishment. Notably, considering that when designing text (especially text that needs an underlay), designers often treat the text and its underlay as a single unified element. To better reflect the practical value of our work, "Banner" refers to elements where Intersection over Union (IoU) or IoD between the text and its underlay is greater than 0.95. We evaluate all baselines based on the above setting. Additionally, we create an extra hard split for each dataset. This hard split is selected from the test and validation sets if any of the following conditions are satisfied: (1) one region is nested within another, (2) a parallel relationship, and (3) the number of elements exceeding four.

### 4.2 BASELINES

(1) RALF (Horita et al., 2024) uses retrieval augmentation to address data scarcity. Unlike the original setting that limited PKU to 10 elements, we extend this to 20 for more complex layouts and fairer comparison. (2) LayoutPrompter (Lin et al., 2024) uses dynamic exemplar selection to avoid LLM training. We adopt GPT-3.5 turbo since the original GPT-3 (text-davinci-003) is no longer available. (3) PosterO (Hsu & Peng, 2025) is a training-free method built on Llama 3.1-8B (Dubey et al., 2024). (4) PosterLlama (Seol et al., 2024) is based on CodeLlama-7B (Roziere et al., 2023).

### 4.3 EVALUATION METRICS

Following the evaluation metrics from previous works (Zhou et al., 2022; Hsu et al., 2023), we apply five metrics. Additionally, we refine the overlap metric to ensure a more reasonable evaluation.

| | Method | $\Delta$Val↓ | Ove↓ | FD↓ | Rea↓ | Occ↓ |
|---|---|---|---|---|---|---|
| PKU annotated | Real Data | 0.0000 (± 0.0000) | 0.0047 (± 0.0000) | - | 0.1673 (± 0.0000) | 0.0387 (± 0.0000) |
| | RALF | **0.0000** (± **0.0000**) | 0.1740 (± 0.0031) | 26.7978 (± 0.2282) | 0.1728 (± 0.0003) | 0.0639 (± 0.0010) |
| | LayoutPrompter | 0.0632 (± 0.0000) | 0.0170 (± 0.0000) | 16.7438 (± 0.0000) | 0.1883 (± 0.0000) | 0.1530 (± 0.0000) |
| | PosterO | 0.0078 (± 0.0024) | 0.1444 (± 0.0043) | 15.3882 (± 0.4792) | 0.1688 (± 0.0014) | **0.0601** (± **0.0017**) |
| | PosterLlama-7B | 0.0007 (± 0.0002) | 0.0318 (± 0.0021) | 5.9256 (± 0.1448) | 0.1727 (± 0.0003) | 0.0659 (± 0.0006) |
| | Qwen2.5VL-7B | 0.0007 (± 0.0003) | 0.0202 (± 0.0010) | 3.6874 (± 0.1903) | **0.1686** (± **0.0003**) | 0.0650 (± 0.0008) |
| | InternVL2.5-8B | 0.0050 (± 0.0014) | 0.0323 (± 0.0014) | 4.3106 (± 0.2717) | 0.1717 (± 0.0002) | 0.0661 (± 0.0025) |
| | ReLayout$^{\diamondsuit}$ | 0.0006 (± 0.0002)↓ | 0.0253 (± 0.0018)↓ | 4.8968 (± 0.0891)↓ | 0.1756 (± 0.0013) | 0.0647 (± 0.0017)↓ |
| | ReLayout$^{\dagger}$ | 0.0005 (± 0.0004)↓ | 0.0152 (± 0.0016)↓ | **2.3931** (± **0.1016**)↓ | 0.1687 (± 0.0004) | 0.0638 (± 0.0011)↓ |
| | ReLayout$^{\heartsuit}$ | 0.0004 (± 0.0005)↓ | **0.0109** (± **0.0001**)↓ | 3.4615 (± 0.1304)↓ | 0.1727 (± 0.0005) | 0.0637 (± 0.0002)↓ |
| CGL annotated | Real Data | 0.0000 (± 0.0000) | 0.0100 (± 0.0000) | - | 0.1758 (± 0.0000) | 0.0540 (± 0.0000) |
| | RALF | 0.0213 (± 0.0001) | 0.0478 (± 0.0010) | **1.7152** (± **0.0557**) | **0.1760** (± **0.0002**) | **0.0518** (± **0.0001**) |
| | LayoutPrompter | 0.0184 (± 0.0000) | 0.0124 (± 0.0000) | 9.3699 (± 0.0000) | 0.1932 (± 0.0000) | 0.1313 (± 0.0000) |
| | PosterO | 0.0025 (± 0.0007) | 0.1309 (± 0.0006) | 13.8778 (± 0.2224) | 0.1761 (± 0.0001) | 0.0545 (± 0.0006) |
| | PosterLlama-7B | 0.0017 (± 0.0004) | 0.0183 (± 0.0013) | 7.1272 (± 0.0236) | 0.1799 (± 0.0003) | 0.0747 (± 0.0005) |
| | Qwen2.5VL-7B7 | 0.0084 (± 0.0013) | 0.0174 (± 0.0006) | 4.3242 (± 0.0535) | 0.1766 (± 0.0001) | 0.0551 (± 0.0005) |
| | InternVL2.5-8B | 0.0098 (± 0.0010) | 0.0195 (± 0.0009) | 4.3051 (± 0.0629) | 0.1765 (± 0.0002) | 0.0588 (± 0.0007) |
| | ReLayout$^{\diamondsuit}$ | **0.0014** (± **0.0006**)↓ | 0.0161 (± 0.0009)↓ | 5.9611 (± 0.0331)↓ | 0.1794 (± 0.0003) | 0.0691 (± 0.0011)↓ |
| | ReLayout$^{\dagger}$ | 0.0039 (± 0.0004)↓ | 0.0147 (± 0.0011)↓ | 3.3870 (± 0.1013)↓ | 0.1768 (± 0.0003) | 0.0545 (± 0.0003)↓ |
| | ReLayout$^{\heartsuit}$ | 0.0023 (± 0.0001)↓ | **0.0117** (± **0.0006**)↓ | 3.1917 (± 0.0215)↓ | **0.1760** (± **0.0001**)↓ | 0.0580 (± 0.0004)↓ |

Table 1: Performance comparison of the C $\rightarrow$ S + P layout generation task on the hard split of PKU and CGL datasets. The best result is highlighted in bold, the second-best result is underlined. $\diamondsuit$, $\dagger$, and $\heartsuit$ represent our method applied with PosterLlama, Qwen2.5VL, and InternVL2.5, respectively. ReLayout mentioned below refers to the row highlighted in red.

| | LayoutPrompter | PosterO | RALF | PosterLlama | InternVL | ReLayout |
|---|---|---|---|---|---|---|
| $P_{use}$ | 36.0% | 44.7% | 50.3% | 71.3% | 78.3% | **91.0%** |
| $P_{best}$ | 0.7% | 7.0% | 6.3% | 11.0% | 10.3% | **64.7%** |

Table 2: User study on structural evaluation. If $P_{use}$ is below 50%, the method is deemed unusable and will be excluded from the diversity evaluation in the user study to save human resources.

| | RALF | PosterLlama | InternVL | ReLayout |
|---|---|---|---|---|
| Score | 41 | 47 | 36 | **56** |
| **cnt** | (18, 23, 9) | (14, 25, 11) | (21, 22, 7) | **(11, 22, 17)** |

Table 3: User study on diversity evaluation. **cnt** denotes the number of generated layout groups (across three different seeds) that exhibit 1, 2, and 3 distinct styles, respectively.

**Graphic metrics:** These metrics evaluate the graphic quality of the layout without considering the canvas. Validity (Val) measures the proportion of elements larger than 0.1% of the canvas; other metrics are computed on these valid elements. Due to the presence of small elements like embellishments in CGL, we use $\Delta$Val for evaluation. In previous works, Overlap (Ove) is the average IoU across all element pairs. However, it has a key limitation: if a layout includes one fully overlapping pair amid many non-overlapping ones, the metric may not reflect poor quality. In reality, heavy overlap between elements is directly seen as a failure. Therefore, we use the maximum IoU to evaluate the generated layouts. We calculate Fréchet Distance (FD) in the feature space derived from bounding boxes and categories to evaluate overall layout quality.

**Content metrics:** These metrics assess harmony between the generated layout and the canvas. Occlusion (Occ) calculates the pixel coverage ratio of layout elements over saliency maps. Readability (Rea) evaluates text clarity using average pixel gradients, where lower is better.

**User study:** In the layout generation field, the current metrics are insufficient to fully evaluate the quality of a layout. Therefore, we conduct user studies. For user studies, we use images from the PKU dataset and invite 6 professional designers as evaluators. For each image, layouts are generated using five different methods and presented simultaneously in a shuffled order, with model names not perceived by users. In terms of structure, 300 images are randomly selected. Users assess each row of results based on two criteria: (1) identify all layouts that meet basic usability standards (e.g., no overlap, no occlusion), denoted as $P_{use}$; and (2) select the single best layout according to professional design principles, considering appropriate margin, relative size, distance from products, reading order, visual priority and overall visual harmony, denoted as $P_{best}$.

In terms of diversity, 50 images are randomly selected. Due to poor usability, LayoutPrompter and PosterO are excluded, leaving four methods, each run with three different seeds (0, 1, 2). Users are

instructed to evaluate diversity based on differences in relative position (e.g., alignment) and text size—any variation in either aspect is considered a distinct style. Diversity is scored as 0 (one style), 1 (two styles), or 2 (three styles).

## 4.4 MAIN RESULTS

Since designers usually design elements first before arranging the overall layout, our experiments primarily focus on generating the positions and sizes of elements based on given categories that each model can support as input.

**Quantitative comparison:** Table 1 presents a comparison of different methods on the hard split of PKU and CGL datasets. Our method improves the performance of various MLLMs, with ReLayout$^\heartsuit$ achieving better results across all metrics. Specifically, on the PKU dataset, it demonstrates the best performance on most metrics, with a particularly notable improvement in the Ove metric. On the CGL dataset, while it does not achieve the best performance across all metrics, it consistently outperforms others on the Ove metric. These improvements are attributed to the annotations margin property of our relation-CoT and resampling strategy that effectively balances the dataset. It demonstrates that our method is better at generating more structured layouts. Although RALF and PosterO perform well on the Occ metrics, their higher Ove score significantly compromises

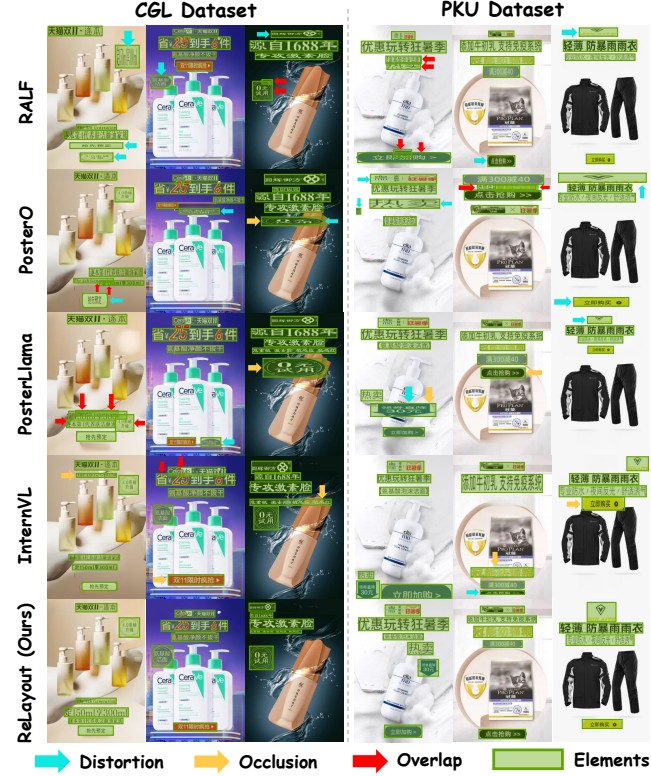

Figure 6: Qualitative comparison on the PKU and CGL datasets. Baselines layouts show noticeable errors, while ours meet basic needs and better align with human aesthetics in margin and arrangement.

their practical usability, as also illustrated in Figure 6. Plus, Table 2 shows that ReLayout performs significantly better in aligning with human aesthetic preferences. Furthermore, Table 3 demonstrates that ReLayout also achieves the highest diversity, exhibiting a greater number and variety of distinct layout styles under different seed settings.

**Qualitative comparison:** Figure 6 visualizes the generated layouts, providing a comparison across different methods. It can be observed that, apart from the obvious errors marked in Figure 6, other methods also fall

| | Region | Saliency | Resample | ΔVal↓ | Ove↓ | FD↓ | Rea↓ | Occ↓ |
|---|---|---|---|---|---|---|---|---|
| V0 | - | - | - | 0.0025 | 0.0153 | 8.7960 | **0.1746** | 0.0821 |
| V1 | ✓ | - | - | 0.0021 | 0.0379 | 12.2290 | 0.1967 | 0.1188 |
| V2 | ✓ | ✓ | - | 0.0014 | 0.0150 | 7.3406 | 0.1769 | 0.0754 |
| V3 | ✓ | ✓ | ✓ | **0.0002** | **0.0097** | **4.9403** | 0.1755 | **0.0752** |

Table 4: Ablation study on the hard split of PKU dataset.

short in controlling element aspect ratio, element spacing, selecting layout arrangements, and achieving overall harmony. In contrast, ReLayout aligns more closely with human aesthetic preferences. Additionally, our method excels at generating diverse layouts and handling layout generation under various conditions, as shown in Figure 11 in the supplementary material.

## 4.5 ABLATION STUDY AND ANALYSIS

**Effect of Each Module:** To simplify the setup, the ablation study uses a training set with only a single main condition from PKU: generating position and size given the category, text, and aspect ratio. As shown in Table 4, V1 adds only region annotations, V2 builds on V1 by incorporating

| $\theta$ | $\Delta$Val↓ | Ove↓ | FD↓ | Rea↓ | Occ↓ |
|---|---|---|---|---|---|
| 3 | 0.0009 | 0.0121 | 4.7920 | 0.1742 | **0.0633** |
| 6 | **0.0004** | **0.0109** | **3.4615** | **0.1727** | 0.0637 |
| 10 | 0.0082 | 0.0168 | 5.0655 | 0.1791 | 0.0703 |
| 100 | 0.0075 | 0.0146 | 4.9287 | 0.1783 | 0.0808 |

Table 5: Hyperparameter analysis on the hard split of PKU dataset.

| $K$ | $\Delta$Val↓ | Ove↓ | FD↓ | Rea↓ | Occ↓ | Score |
|---|---|---|---|---|---|---|
| 3 | 0.0012 | 0.0223 | 5.0017 | 0.1800 | 0.0677 | 40 |
| 6 | **0.0004** | 0.0141 | 4.4091 | 0.1796 | 0.0640 | 48 |
| 8 | **0.0004** | **0.0109** | **3.4615** | 0.1727 | **0.0637** | 56 |
| 10 | 0.0009 | 0.0183 | 4.9004 | **0.1724** | 0.0639 | 57 |

Table 6: Cluster number analysis on the hard split of PKU dataset.

saliency annotations, and V3 builds on V2 by additionally introducing a layout prototype rebalance sampler, several observations can be made. First, V1 demonstrates relatively poor overall metrics, likely due to the model focusing on the structure of elements while neglecting salient objects. Since no structure-related metrics, this effect cannot be quantified and must be analyzed via visualization in the supplementary material. Second, V2 shows that FD improves compared to V0. Compared to the Region-only setting, all metrics show an upward trend, particularly the Occ metric, which demonstrates the importance of Saliency in content-aware tasks. Third, V3 achieves the best results. Notably, the improvements in Ove and FD are particularly significant.

In addition, we conducted detailed ablation studies on the margin attribute and the feature vectors. The results are presented in Table 7. It can be clearly observed that the margin attribute makes a significant contribution to mitigating the element overlap problem. Simultaneously, the ablation of the feature vectors suggests that the region feature vector may slightly degrade the Rea metric. We hypothesize that this phenomenon occurs because the region feature tends to place elements in areas that are semantically richer, which might incidentally contain stronger texture features, thus leading to a minor reduction in Rea.

**Hyperparameter Analysis:** We analyze the hyperparameter $\theta$ on the hard split of PKU dataset, as shown in Table 5. The results show that $\theta = 6$ yields the best performance. A smaller $\theta$ better preserves the original distribution, causing rare layout prototypes to be treated as noise. In contrast, a larger $\theta$ overly balances the distribution, leading to repeated learning of rare prototypes and reduced perfor-

|  | $\Delta$Val↓ | Ove↓ | FD↓ | Rea↓ | Occ↓ |
|---|---|---|---|---|---|
| Ours | **0.0004** | **0.0109** | **3.4615** | 0.1727 | **0.0637** |
| w/o margin | 0.0007 | 0.0301 | 4.1017 | 0.1733 | 0.0638 |
| w/o $\mathbf{f}_i^s$ | 0.0011 | 0.0181 | 3.5011 | 0.1727 | 0.0644 |
| w/o $\mathbf{f}_i^r$ | 0.0007 | 0.0166 | 3.5772 | **0.1723** | 0.0639 |
| w/o $\mathbf{f}_i^e$ | 0.0009 | 0.0207 | 3.8439 | 0.1728 | 0.0638 |

Table 7: Ablation study on margin attributes and feature vectors on the PKU dataset.

mance. We also analyze the hyperparameter $K$ (the number of clusters) on the hard split of the PKU dataset. As shown in Table 6, $K = 3$ results in poorly discriminative prototypes, which fail to fully capture the underlying diversity of layout patterns. Although a setting of $K = 10$ achieves the highest diversity score, its overall quantitative metrics are significantly worse compared to $K = 8$. Therefore, setting $K = 8$ proves to be the optimal choice.

## 5 CONCLUSION

In this work, we study content-aware layout generation tasks and address the issue in LLM-based methods where the relationships between elements have not been considered. We propose a novel method ReLayout, which consists of two modules. First, we enhance the model's understanding of relationships by incorporating explicit relationship annotations, framed from the perspective of CoT. Second, we utilize relation annotations to cluster the dataset and adjust its distribution, thereby enhancing the quality of generated layouts. Moreover, extensive experiments validate the effectiveness of our method, particularly in visualization.

Furthermore, we identify some limitations in ReLayout. First, while current metrics can effectively identify obviously inadequate layouts, they lack the capability to evaluate layout suitability for real-world application scenarios. Second, the processing performance is constrained by the limitations of multi-resolution datasets, resulting in suboptimal results. Third, the model demonstrates difficulty in handling a large number of input elements. Furthermore, when earlier regions occupy the entire available canvas, subsequent predictions are likely to overlap.

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

# A IMPLEMENTATION DETAILS

## A.1 TRAINING SETS SETTING

Our method is applied to three MLLMs: PosterLlama-7B (Seol et al., 2024), Qwen2.5VL-7B (Team, 2025), and InternVL2.5-8B (Chen et al., 2024). Each experiment is conducted on 8 A800 GPUs. We follow their default settings for training and inference.

When training on the PKU dataset, we adopt a diverse set of seven conditional settings to enable flexible and robust layout generation under various input combinations. These include: `cate_text_eimg_to_size_pos`, where the model predicts element size and position based on category, textual description, and element images; `cond_cate_text_eimg_size_to_pos`, which infers only the position of elements given their category, text, element images, and sizes; `text_eimg_recover_mask`; `text_eimg_refinement`, which refines initial coarse predictions based on multimodal input; `cate_text_eimg_ar_to_size_pos`, where aspect ratio is additionally considered alongside category, text, and element images to predict size and position; `text_eimg_completion`, which completes partially missing layout information; and `cate_text_to_size_pos`, a relatively simpler setting that predicts element size and position based only on category and text, without element images input. In contrast, when training on the CGL dataset, we use only one condition: `cate_text_eimg_ar_to_size_pos`, which focuses on predicting element size and position from a complete set of multimodal inputs including category, text, element images, and aspect ratio, aligning with the structure and requirements of the CGL dataset.

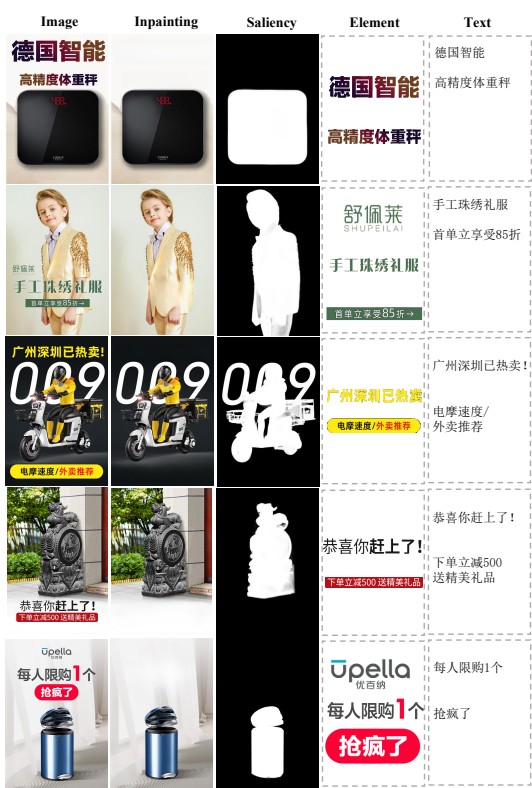

Figure 7: Visualization of Data Preprocessing.

## A.2 DATA PREPROCESSING

We use LaMa (Suvorov et al., 2022) for inpainting and PaddleOCR for text extraction on both PKU and CGL datasets as shown in Figure 7. It can be observed that our inpainting results effectively remove elements, and the text extraction is precise. The saliency map is crucial for Algorithm 1. We employ BASNet (Qin et al., 2019) and ISNet (Zhang et al., 2022) to obtain the saliency map. Figure 7 illustrates the original image, inpainting results, the saliency map generated by ISNet, element extraction, and text extraction, respectively.

We define the algorithm for the Salient box as shown in Algorithm 1, which outlines the process of detecting salient regions in an image. The algorithm iteratively searches for the most prominent rectangles based on intensity thresholds and overlap constraints, ensuring that the selected regions maximize their saliency scores while minimizing redundancy. This approach effectively identifies the most important areas in the image.

Algorithm 2 determines whether the dominant layout direction is "row" or "column" by analyzing the spatial distribution of the bounding boxes $\mathcal{B}$. It first projects the boxes onto the x-axis and y-axis to analyze horizontal and vertical alignment separately. For each axis, it groups overlapping boxes using a specified IoD (Yu et al., 2020) threshold $\phi$. If the layout forms a single group along one axis

---

**Algorithm 1** Salient Region Detection

---

**Input:** Image $I$, maximum rectangles $N$, search step $s$
**Output:** List of salient rectangles $R$
1: $B \leftarrow \mathbb{I}(I > \tau)$ where $\tau$ is intensity threshold
2: $I_{\text{white}}, I_{\text{black}} \leftarrow \text{ComputeIntegralImage}(B)$
3: $V \leftarrow \mathbf{0}, I_{\text{v}} \leftarrow \text{ComputeIntegralImage}(V)$
4: $R \leftarrow \emptyset$
5: **for** $i \leftarrow 1$ to $N$ **do**
6:     $max\_score \leftarrow 0, best\_rect \leftarrow$ null
7:     **for** $r \in \text{SearchSpace}(s)$ **do**
8:         **if** $\exists a \in R : \text{Overlap}(a, r)$ **then**
9:             **continue**
10:         **end if**
11:         $w \leftarrow \text{RectSum}(I_{\text{white}}, r) - \text{RectSum}(I_{\text{v}}, r)$
12:         $b \leftarrow \text{RectSum}(I_{\text{black}}, r)$
13:         $score \leftarrow \text{ComputeScore}(w, b, r)$
14:         **if** $score > max\_score$ **then**
15:             $max\_score, best\_rect \leftarrow score, r$
16:         **end if**
17:     **end for**
18:     $R \leftarrow R \cup \{best\_rect\}$, update $V$ and $I_{\text{v}}$
19: **end for**
20: **return** $R$

---

**Algorithm 2** Estimate Layout Direction

---

**Input:** Bounding boxes $\mathcal{B}$, overlap threshold $\phi$.
**Output:** (Direction, $G$)
1: $L_x \leftarrow$ project bounding boxes $\mathcal{B}$ to x-axis;
2: $L_y \leftarrow$ project bounding boxes $\mathcal{B}$ to y-axis;
3: $G_x \leftarrow \text{GroupByOverlap}(L_x, \phi)$;
4: $G_y \leftarrow \text{GroupByOverlap}(L_y, \phi)$;
5: **if** $|G_x| = 1$ AND $|G_y| > 1$ **then**
6:     **return** ("column", $G_y$)
7: **else if** $|G_y| = 1$ AND $|G_x| > 1$ **then**
8:     **return** ("row", $G_x$)
9: **else**
10:     $V_x \leftarrow \text{ComputeGroupVariance}(G_x)$
11:     $V_y \leftarrow \text{ComputeGroupVariance}(G_y)$
12:     **if** $V_x \leq V_y$ **then return** ("row", $G_x$)
13:     **else**
14:         **return** ("column", $G_y$)
15:     **end if**
16: **end if**
17: **function** GROUPBYOVERLAP$(L, \phi)$
18:     $edges \leftarrow \emptyset$
19:     **for** each pair $(i, j)$ in $L$ **do**
20:         **if** $\text{IoD}(L[i], L[j]) \geq \phi$ **then**
21:             $edges \leftarrow edges \cup \{(i, j)\}$
22:         **end if**
23:     **end for**
24:     $groups \leftarrow \text{FindConnectedComponents}(edges)$
25:     **return** $groups$
26: **end function**

---

but multiple groups along the other, it selects the direction with more structural variation (i.e., more groups). If both axes have multiple groups, it compares the intra-group variance in each direction

and chooses the one with smaller variance as the dominant direction. The algorithm returns both the inferred direction ("row" or "column") and the corresponding grouped structure.

### A.3 HYPERPARAMETER SETTING

We set the feature clustering parameters $\alpha$, $\beta$, and $\gamma$ in the layout prototype rebalance sampler to 2, 1, and 10, respectively.

## B INPUT-OUTPUT PROMPT EXAMPLE

Here, we present an example prompt for specifying a layout generation task. We illustrate a C $\rightarrow$ S + P example that provides an overview of all tasks.

## C MORE QUANTITATIVE RESULTS

### C.1 OUT-OF-DOMAIN GENERALIZATION

To verify the generalization of our method, we conduct experiments using PKU as the training set and testing on CGL, and vice versa. As shown in Table 9, our method outperforms the current SOTA PosterLlama on most metrics. This demonstrates that ReLayout adapts well to real-world scenarios and demonstrates strong generalization performance.

### C.2 RELAYOUT ON TEST SPLIT

In addition to evaluating the effectiveness of our method on the hard split, which focuses on more challenging layout scenarios, we further conduct experiments on the test split of the PKU and CGL datasets. This comprehensive evaluation aims to assess the generalization ability and robustness of our approach across different task settings. From the results shown in Table 10, our method achieves consistently strong performance across different dataset splits, demonstrating that the proposed layout relation-CoT construction and layout prototype rebalance sampler strategy enables our model to adapt well to varying data partitions.

### C.3 LAYOUTPROMPTER BASED ON GPT4O

We follow prior work by first evaluating the performance of GPT-3.5-turbo using the same experimental settings. In addition, we extend the evaluation to include GPT-4o on the PKU-test set after making the necessary adjustments to ensure compatibility with our codebase. As shown in Table 11, GPT-4o achieves better results compared to GPT-3.5-turbo, demonstrating its stronger reasoning and layout generation capabilities. However, despite these improvements, our proposed method still outperforms both GPT-3.5-turbo and GPT-4o, highlighting the effectiveness of our approach in modeling layout relations more accurately.

### C.4 SCALING LAW ANALYSIS

Based on the results in Table 12, a clear scaling trend emerges: the 8B model consistently and substantially outperforms the 4B model across all metrics and both test splits. This indicates that increasing model size yields significant performance improvements, particularly on the hard split of PKU dataset, where the 8B model shows a marked advantage. The consistent gains across metrics and splits suggest that scaling up model size is an effective strategy for improving both capability and robustness.

### C.5 COMPUTATIONAL COST AND RUNNING TIME

Table 13 reports the inference performance of different methods, evaluated on an NVIDIA L20 GPU using a single-sample test setting for fair comparison. Compared to PosterLlama, our method achieves better performance while maintaining nearly the same inference time, indicating that the improvements in layout quality do not come at the cost of efficiency. Although our method incurs

---

**Input**

---

**Task Definition**

Given canvas image: `<image>`

Given elements Images: `<image><image><image><image>`

Please generate a layout based on the given information for the given canvas image. You need to ensure that the generated layout (in html format) looks realistic, with elements well aligned and avoiding unnecessary overlap.

**Task Description**

layout generation with given class.

**Layout Domain**

advertisement layout.

**Input html**

```html
<div class="canvas" style="left:0px;_top:0px;width:683px;height:999px">
  <div class="logo" style="left:<X1>px;top:<Y1>px;width:<W>px;height:<H>px">
  </div>
  <div class="text" style="left:<X1>px;top:<Y1>px;width:<W>px;height:<H>px"
  text="half"></div>
  <div class="text" style="left:<X1>px;top:<Y1>px;width:<W>px;height:<H>px"
  text="announce"></div>
  <div class="banner" style="left:<X1>px;top:<Y1>px;width:<W>px;height:<H>px"
  text="size"></div>
</div>
```

---

**Output html**

```html
<div class="canvas" style="left:0px;top:0px;width:683px;height:999px">
  <div class="relation_salient">
    <div class="salient" style="left:200px;top:450px;width:301px;height:450px">
    </div>
  </div>
  <div class="layout">
    <div class="relation_region" style="flex-direction:row;align-items:stretch;
left:32px;top:37px;width:631px;height:409px">
    <div class="relation_region" style="flex-direction:column;align-items:center;
left:32px;top:37px;width:433px;height:331px">
      <div class="logo" style="margin-top:0px;left:225px;top:37px;width:240px;
height:104px"></div>
      <div class="text" style="margin-top:69px;left:32px;top:210px;width:328px;
height:83px"></div>
      <div class="banner" style="margin-top:17px;left:51px;top:310px;
width:293px;height:58px"></div>
    </div>
    <div class="text" style="margin-left:-92px;left:373px;top:181px;
width:290px;height:265px"></div>
    </div>
  </div>
</div>
```

---

Table 8: Prompt example

---

slightly higher FLOPs, this is primarily due to the generation of longer outputs, which reflects the richer relational information being modeled. As for LayoutPrompter, we do not report its inference statistics because it relies on external calls to GPT API.

| Train | Test | Method | ΔVal↓ | Ove↓ | FD↓ | Rea↓ | Occ↓ |
|---|---|---|---|---|---|---|---|
| PKU | CGL-hard | PosterLlama | 0.0225 (± 0.0001) | 0.0311 (± 0.0004) | 6.5679 (± 0.1091) | 0.1758 (± 0.0004) | 0.0688 (± 0.0010) |
| | | ReLayout (Ours) | **0.0167** (± **0.0001**) | **0.0100** (± **0.0004**) | **4.4413** (± **0.0136**) | **0.1715** (± **0.0001**) | **0.0631** (± **0.0002**) |
| CGL | PKU-hard | PosterLlama | 0.0019 (± 0.0007) | 0.0205 (± 0.0035) | 7.1093 (± 0.2796) | **0.1726** (± **0.0010**) | 0.0694 (± 0.0016) |
| | | ReLayout (Ours) | **0.0010** (± **0.0005**) | **0.0120** (± **0.0023**) | **5.9011** (± **0.1120**) | 0.1730 (± 0.0012) | **0.0660** (± **0.0009**) |

Table 9: Cross-dataset evaluation on PKU and CGL datasets.

| Method | Test Split | | | | |
|---|---|---|---|---|---|
| | Graphic | | | Content | |
| | ΔVal↓ | Ove↓ | FD↓ | Rea↓ | Occ↓ |
| **PKU Annotated Dataset** | | | | | |
| Real Data | 0.0000 (± 0.0000) | 0.0035 (± 0.0000) | - | 0.1545 (± 0.0000) | 0.0639 (± 0.0000) |
| LayoutPrompter (Lin et al., 2024) | 0.0015 (± 0.0000) | 0.0090 (± 0.0000) | 8.0392 (± 0.0000) | 0.1683 (± 0.0000) | 0.1452 (± 0.0000) |
| RALF (Horita et al., 2024) | **0.0000** (± **0.0000**) | 0.0915 (± 0.0023) | 15.5497 (± 0.1499) | 0.1617 (± 0.0005) | 0.0866 (± 0.0024) |
| PosterLlama-T (Seol et al., 2024) | 0.0002 (± 0.0003) | 0.0211 (± 0.0018) | 3.5318 (± 0.2160) | 0.1612 (± 0.0002) | 0.0863 (± 0.0019) |
| InternVL2.5-8B (Chen et al., 2024) | 0.0054 (± 0.0009) | 0.0175 (± 0.0004) | 2.6175 (± 0.0905) | **0.1588** (± **0.0003**) | 0.0885 (± 0.0016) |
| ReLayout (Ours) | 0.0001 (± 0.0002) | **0.0086** (± **0.0011**) | **1.7865** (± **0.1195**) | 0.1600 (± 0.0004) | **0.0857** (± **0.0010**) |
| **CGL Annotated Dataset** | | | | | |
| Real Data | 0.0000 (± 0.0000) | 0.0060 (± 0.0000) | - | 0.1654 (± 0.0000) | 0.0771 (± 0.0000) |
| LayoutPrompter (Lin et al., 2024) | 0.0125 (± 0.0000) | 0.0094 (± 0.0000) | 6.7951 (± 0.0000) | 0.1787 (± 0.0000) | 0.1510 (± 0.0000) |
| RALF (Horita et al., 2024) | 0.0147 (± 0.0001) | 0.0283 (± 0.0007) | **0.9277** (± **0.0312**) | 0.1649 (± 0.0002) | **0.0744** (± **0.0001**) |
| PosterLlama-T (Seol et al., 2024) | 0.0012 (± 0.0003) | 0.0102 (± 0.0010) | 4.4151 (± 0.0129) | 0.1674 (± 0.0001) | 0.0931 (± 0.0004) |
| InternVL-2.5-8B (Chen et al., 2024) | 0.0062 (± 0.0005) | 0.0114 (± 0.0007) | 2.8395 (± 0.1031) | 0.1649 (± 0.0002) | 0.0796 (± 0.0003) |
| ReLayout (Ours) | **0.0004** (± **0.0002**) | **0.0088** (± **0.0003**) | 1.9311 (± 0.0120) | **0.1648** (± **0.0001**) | 0.0787 (± 0.0001) |

Table 10: Performance comparison of the C → S + P layout generation task on the PKU and CGL datasets. The best result is highlighted in bold, the second-best result is underlined, and the row corresponding to our method is marked in red.

| | ΔVal↓ | Ove↓ | FD↓ | Rea↓ | Occ↓ |
|---|---|---|---|---|---|
| 3.5-turbo | 0.0015 | 0.0090 | 8.0392 | 0.1683 | 0.1452 |
| 4o | 0.0010 | **0.0066** | 7.6730 | 0.1648 | 0.1472 |
| ReLayout | **0.0001** | 0.0086 | **1.7865** | **0.1600** | **0.0857** |

Table 11: LayoutPrompter based on GPT4o.

# D  MORE QUALITATIVE RESULTS

## D.1  SOME SMALL DISCOVERIES

We also observe that ReLayout is capable of capturing and understanding the underlying placement logic associated with certain specific types of advertising promotional terms. As illustrated in the red box in Figure 8, instances of `Quantitative Advertising`, such as "Over 3.49 million units sold!", or `Sale Advertising`, such as "GIFT", are sometimes positioned in ways that partially occlude salient visual elements (e.g., product images or logos). While this may seem suboptimal from a purely visual clarity perspective, such placement actually conforms to common design conventions in advertising, where emphasis on promotional content often takes precedence to attract user attention.

## D.2  VISUALIZATION OF ABLATION STUDIES

We conducted a visual analysis of the ablation studies. As shown in Figure 9, V2 has a more aesthetically pleasing lay-

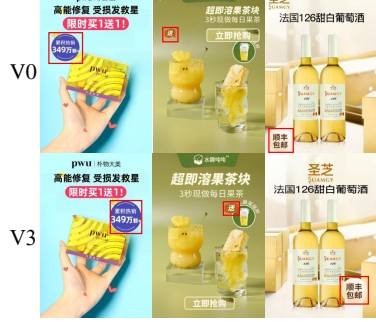

Figure 8: Examples of place promotional terms.

| Params | $\Delta$Val$\downarrow$ | Ove$\downarrow$ | FD$\downarrow$ | Rea$\downarrow$ | Occ$\downarrow$ |
|---|---|---|---|---|---|
| **PKU Test Split** | | | | | |
| 4B | 0.0207 | 0.1732 | 84.4993 | 0.2092 | 0.1774 |
| 8B | **0.0001** | **0.0086** | **1.7865** | **0.1600** | **0.0857** |
| **PKU Hard Split** | | | | | |
| 4B | 0.0204 | 0.1804 | 97.0555 | 0.2114 | 0.1602 |
| 8B | **0.0004** | **0.0109** | **3.4615** | **0.1727** | **0.0637** |

Table 12: Scaling law analysis on PKU dataset.

| Method | Params(B) | Time(s) | FLOPs(T) | $P_{best}$ |
|---|---|---|---|---|
| LayoutPrompter | - | 4.2 | - | 1.3% |
| PosterLlama | 6.9 | 7.5 | 2.2 | 12.0% |
| InternVL | 8.1 | 4.9 | 3.0 | 10.7% |
| ReLayout | 8.1 | 7.5 | 4.0 | **66.3%** |

Table 13: Computational cost and running time analysis.

out compared to V0 and V1, in terms of margin, alignment, and adherence to human visual preferences. Furthermore, as shown in Figure 9, V2 exhibits a clearer understanding of element sizes compared to V1, conveying information more effectively. Additionally, in the last column of the figure, we can see that adding the Resample module in simple layout scenarios does not lead to forgetting.

## D.3 DIVERSE SEED GENERATION

To demonstrate the diversity of layout generation results, we adjust random seeds during the generation process as shown in Figure 10. It can be observed that our method consistently meets the basic layout requirements, such as avoiding overlap and occlusion, across different random seeds. At the same time, the layouts generated by each seed are distinct while aligning with human aesthetic preferences. Notably, variations in random seeds may lead to the enlargement of promotional keywords, emphasis on functional slogans, or adjustments in regional alignment, which further demonstrate the effectiveness and diversity of our approach in layout generation.

## D.4 DIVERSE CONDITIONAL GENERATION

Our method is capable of generating high-quality layouts under various conditions. Specifically, we visualize the generate layouts shown in Figure 11 under the following conditions: (1) Unconditional: Given only the canvas image, the model generates the entire layout; (2) C + S $\rightarrow$ P: Given the category and position of input elements, the model predicts their sizes; (3) Completion: Provided with a partial layout, the model generates a complete one; (4) Refinement: After applying Gaussian perturbations to the layout, the model adjusts it to achieve the optimal arrangement. It can be observed that ReLayout is capable of producing sufficiently high-quality layouts under various user-defined constraints.

## D.5 SPECIFIC RELATION VISUALIZATION

We performed a detailed parse of the relationships in the layouts generated by our method, as shown in Figure 12. It can be seen that our method handles salient positions with high accuracy and organizes the arrangement within regions clearly. The overall layout exhibits a strong sense of hierarchy and logical structure. Additionally, the spacing and proportions between elements are well-controlled, fully reflecting an understanding of human aesthetics. This advantage makes our method particularly effective in complex layout generation tasks.

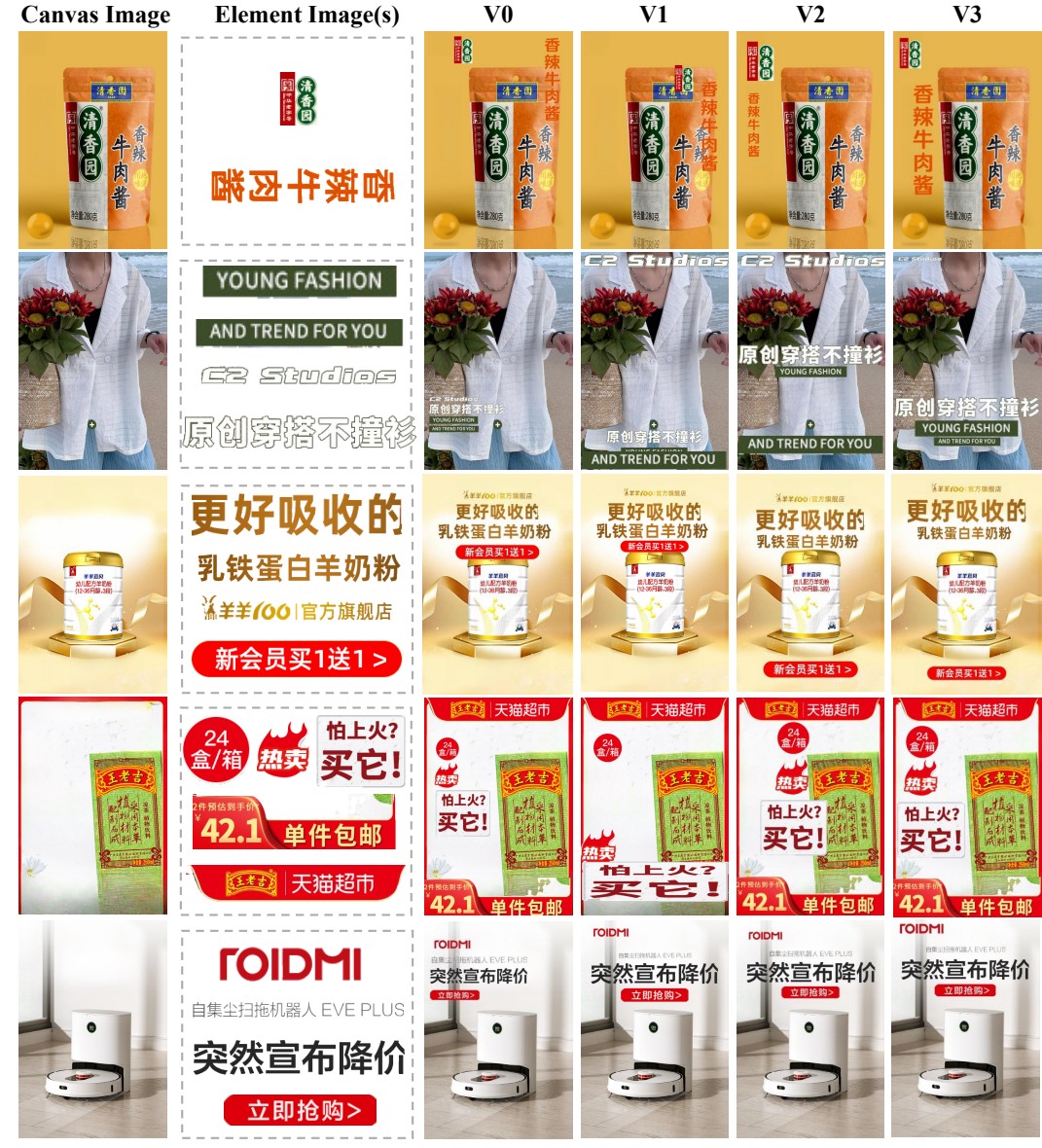

Figure 9: Examples of visualization for ablation studies.

## E    LLMS USAGE

We only utilize LLMs for text polishing purposes. Specifically, LLMs are employed solely to assist with language refinement, including grammar checking, sentence structure optimization, and improving the overall readability and fluency of the manuscript. It is important to emphasize that all core research contributions, including the conceptual framework, methodology, experimental design, data analysis, interpretation of results, and conclusions, are developed entirely by the authors without any involvement of LLMs.

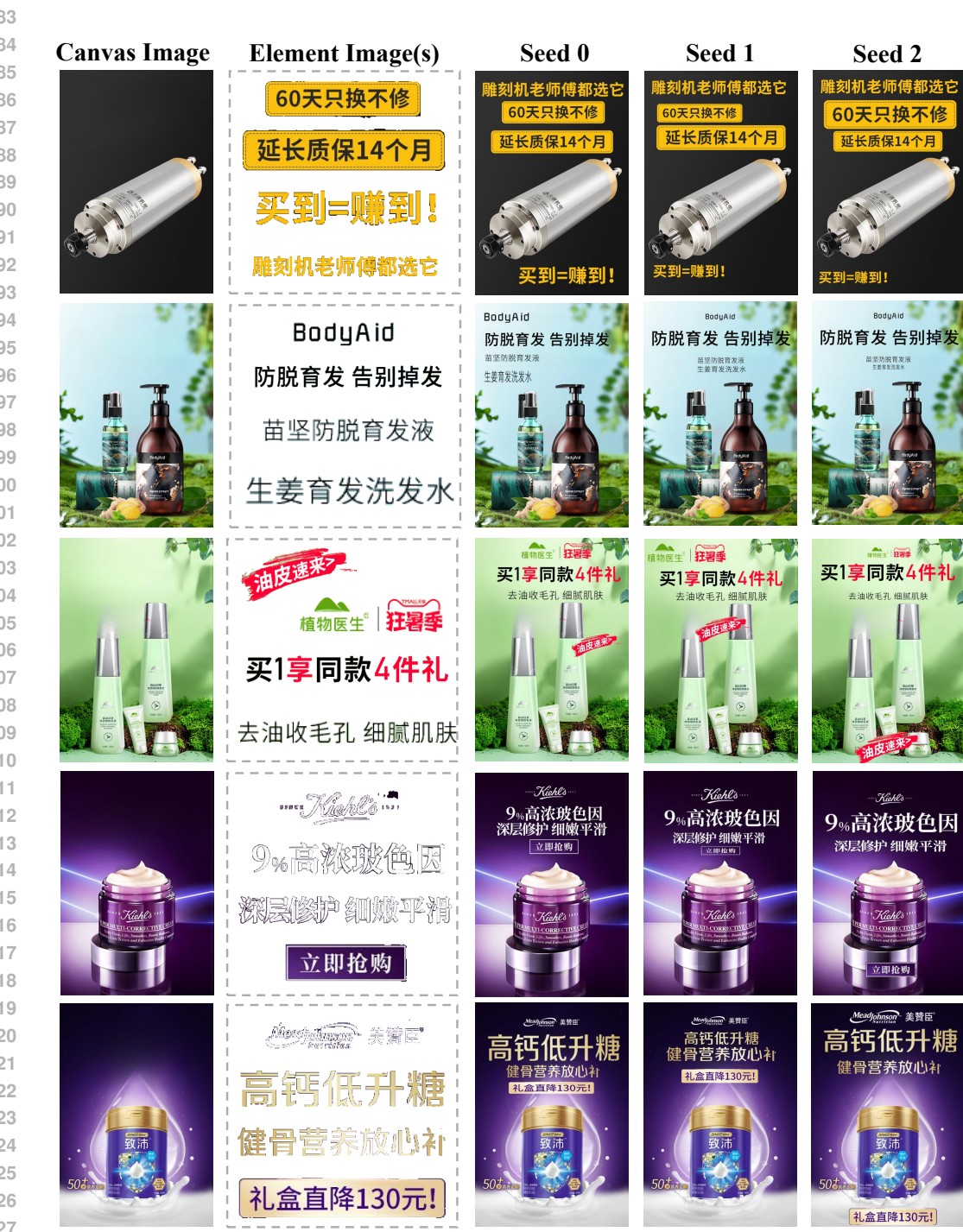

Figure 10: Examples of visualization across diverse seed.

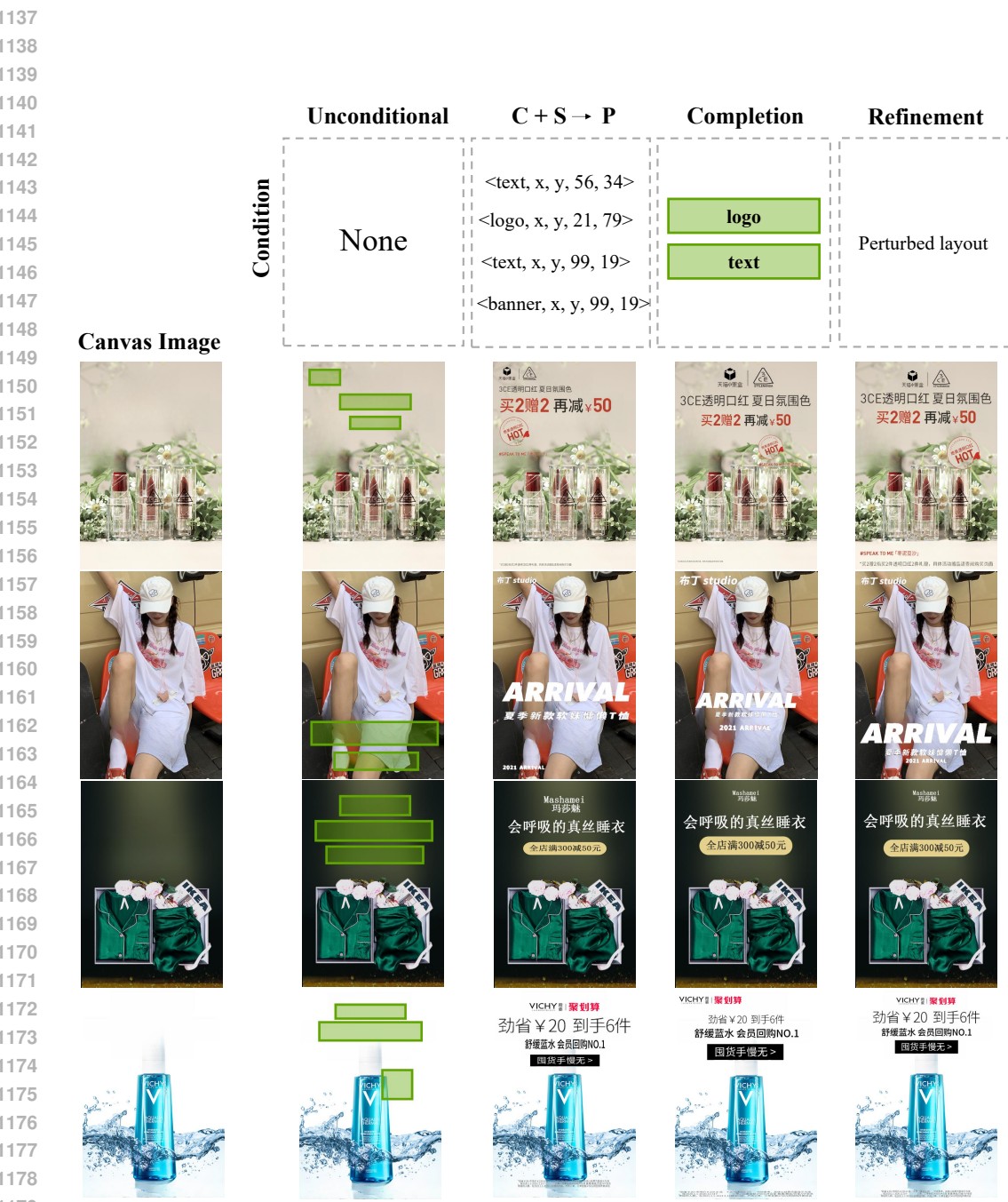

Figure 11: Examples of visualization across diverse conditions.

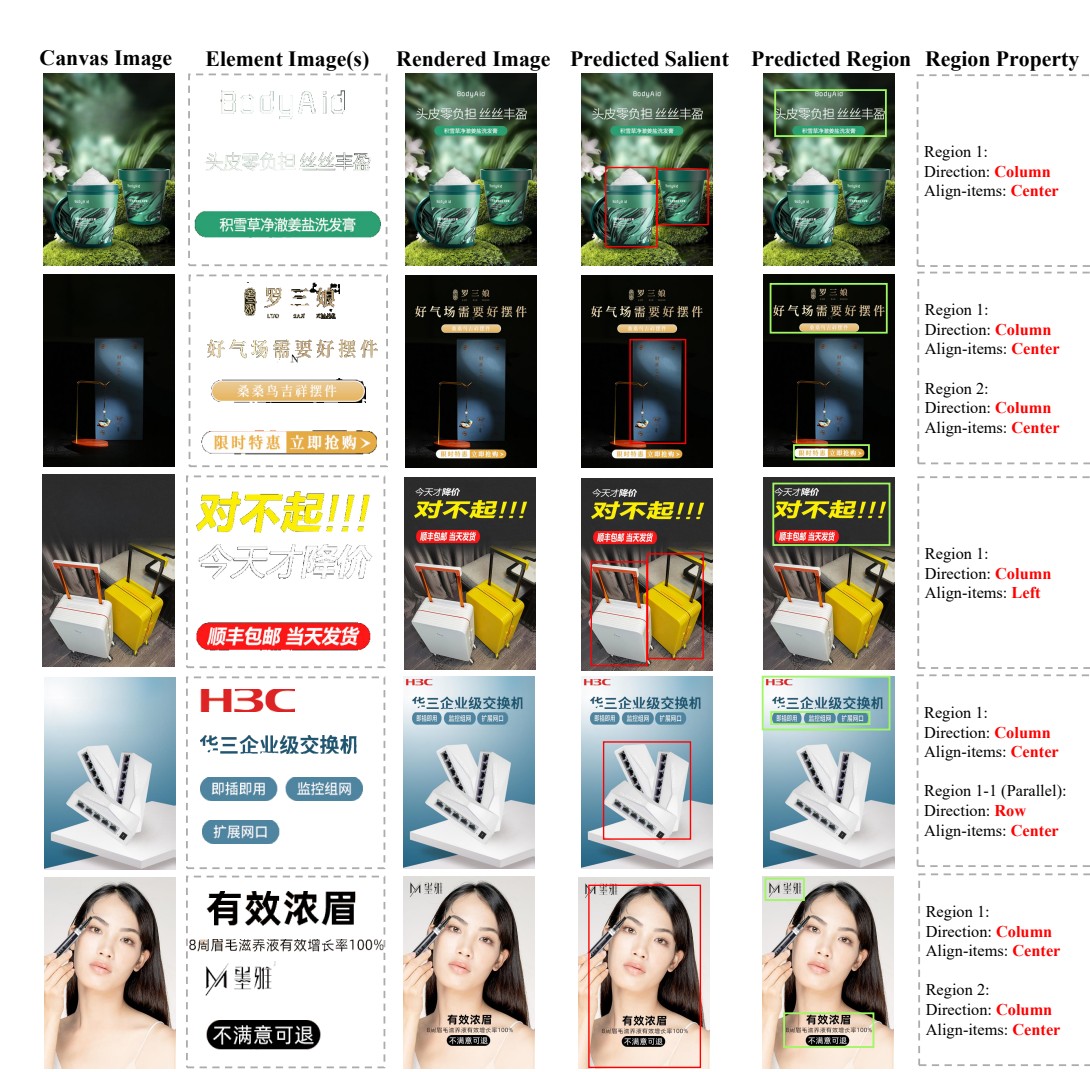

Figure 12: Comprehensive visualization and detailed analysis of the generated output to illustrate relational elements.

