# OpenReview forum: "ReLayout: Integrating Relation Reasoning for Content-aware Layout Generation with Multi-modal Large Language Models"
_ICLR.cc/2026/Conference — Submitted to ICLR 2026_

### Official Review · Reviewer_BWPH · 2025-10-28

**Soundness:** 2
**Presentation:** 2
**Contribution:** 1
**Rating:** 2
**Confidence:** 4

**Summary:**

This paper proposes an approach, ReLayout, to add layout constraints to an existing layout dataset. ReLayout is based on two methods, Layout Relation-CoT Construction method to recover and reconstruct the relationships of the design elements in a hierarchical manner, and Layout prototype Rebalance Sampler to adjust the sample distribution via weighted sampling.

**Strengths:**

1. The paper is generally easy to follow.

2. The relation data added to the dataset may be useful.

**Weaknesses:**

1. Novelty of the proposed idea: This work tries to incorporate some low-level constraints (the relationships among the design elements, such as negative space, alignment, non-overlapping, and saliency) into graphic design layouts. This paper presents it in a way like it is never considered before. In fact, it is largely studied by existing works. For example, [1] explicitly models different types of layout constraints for graphic design; [2] considers the empty space (or negative space) and overlapping among design elements; and [3] considers saliency information in layout generation. This paper does not discuss what existing works have done.

2. Problems with the Layout Relation-CoT construction method. First, I am not sure why the generation of relations is related to CoT. While the relations may be generated in a hierarchical manner, I do not see how it is a step-by-step generation. In fact, I am really confused how this method can even be remotely related to CoT. Second, the recovery of the relations among the design elements is actually rather straightforward. I do not see how this method is novel.

3. Problems with the Layout Prototype Rebalance Sampler. First, by adjusting the sample distribution, I agree that less frequent samples will get sampled more frequently. However, would this also create a negative effect that popular designs become less popular while rare designs become popular? Second, why is the layout prototype rebalance sampler novel? Where is it novel? This is not discussed in the paper.


[1] Peter O’Donovan, Aseem Agarwala, and Aaron Hertzmann. Learning Layouts for Single-Page Graphic Designs. TVCG, 2014.
[2] Xinru Zheng, Xiaotian Qiao, Ying Cao, and Rynson Lau. Content-aware generative modeling of graphic design layouts. TOG (ACM SIGGRAPH), 2019.
[3] Daichi Horita, Naoto Inoue, Kotaro Kikuchi, Kota Yamaguchi, and Kiyoharu Aizawa. Retrieval-Augmented Layout Transformer for Content-Aware Layout Generation. In CVPR, 2024.

**Questions:**

See my comments/questions in the Weaknesses section.

---

> ### Author Response · Authors · 2025-11-23
> **Official Comment to Reviewer BWPH (1 / 2)**
>
> Thank you for your efforts in reviewing our paper. The followings are our responses.
> - - -
> **Weakness 1**: Novelty of the proposed idea.
>
> **Response to weakness 1**:
>
> We thank the reviewer for referencing these foundational works. We agree that design principles like alignment, negative space, and saliency have been introduced.
>
> However, our contribution is not the discovery of these constraints, but a novel **paradigm** to integrate them into MLLMs:
> - **Methodological Shift:** Prior works [1-3] typically treat these as **optimization constraints or loss functions**. In contrast, ReLayout encodes them as **explicit reasoning tokens (Relation-CoT)** within an autoregressive language model. We are the first to model these constraints as a "language" for MLLMs to reason with step-by-step.
> - **Addressing MLLM Limitations:** As noted in our Introduction section, current MLLMs (e.g., PosterLlama [4]) treat layout as a numerical regression task, failing to capture structural logic. Our method explicitly solves this by decomposing layouts into recursive regions via HTML.
> - **Data-Centric Novelty:** Unlike [1-3], we also leverage these relations to construct a Prototype Rebalance Sampler, which explicitly addresses the diversity problem in data-driven generation.
>
> In our revision, we will explicitly cite the works mentioned by the reviewer in Related Work.
>
> [1] Peter O’Donovan, Aseem Agarwala, and Aaron Hertzmann. Learning Layouts for Single-Page Graphic Designs. TVCG, 2014.
>
> [2] Xinru Zheng, Xiaotian Qiao, Ying Cao, and Rynson Lau. Content-aware generative modeling of graphic design layouts. TOG (ACM SIGGRAPH), 2019.
>
> [3] Daichi Horita, Naoto Inoue, Kotaro Kikuchi, Kota Yamaguchi, and Kiyoharu Aizawa. Retrieval-Augmented Layout Transformer for Content-Aware Layout Generation. In CVPR, 2024.
>
> [4] Seol J, Kim S, Yoo J. Posterllama: Bridging design ability of langauge model to contents-aware layout generation[J]. ECCV 2024.
> - - -
> **Weakness 2**: Questionable CoT connection; relation recovery method lacks novelty and is straightforward.
>
> **Response to weakness 2 (1/2)**:
>
> We thank the reviewer for this insightful comment. In our framework, we generalize the concept of CoT from textual reasoning to structural reasoning. Standard LLM CoT decomposes a hard problem into intermediate steps. Similarly, ReLayout decomposes the complex task of generating a layout into a hierarchical dependency chain, rather than directly outputting coordinates.
>
> The step-by-step generation in ReLayout is explicitly modeled through our **recursive region generation** (as detailed in Section 3.2 and Figure 3b):
> - **Step 1 (Saliency):** The model first identifies "Where is the salient objects?"
> - **Step 2 (Global Structure):** It then decides the level-1 regions.
> - **Step 3 (Recursive Refinement):** It recursively decomposes regions into sub-regions or elements.
> - **Step 4 (Final Placement):** Only after this structural chain is established does it generate the specific element coordinates.
> - - -
> **Response to weakness 2 (2/2)**:
>
> We agree that extracting these relations from an existing image (using the heuristic algorithms in Appendix A.2) is straightforward. However, our novelty lies in integrating this structured logic into the generative process of MLLMs, which has not been done before in this domain.
>
> - **The Novelty of Explicit Modeling:** Previous SOTA methods (PosterLlama) treat layout generation as a **numerical regression problem** (predicting $[x, y, w, h]$). They fail to understand relationships, leading to the "structural problems" (overlap, missing parallel) shown in Figure 1(a). By forcing the model to learn these straightforward relations, we transform the task into a **logical structure generation problem**.
> - **Enabling Data Balancing:** This relation construction is the prerequisite for our second novel contribution: the **Layout Prototype Rebalance Sampler** (Section 3.3). Because we recover these relations, we can quantify layout styles and rebalance the training data. This directly solves the diversity problem (Figure 1b) where models collapse into a single mode. Without this straightforward relation extraction, this second novel contribution procedure would be impossible.
>
> All in all, Relation-CoT is not just about the extraction algorithm, but about **shifting the generation paradigm** from predicting coordinates to hierarchical structural reasoning.

---

> ### Author Response · Authors · 2025-11-23
> **Official Comment to Reviewer BWPH (2 / 2)**
>
> **Weakness 3**: Rebalancing may overly suppress popular designs while overlearning rare ones; novelty of the sampler is unclear.
>
> **Response to weakness 3 (1/2)**:
>
> We acknowledge the reviewer's concern that rebalancing can potentially distort the natural data distribution. However, we explicitly addressed this trade-off through the design of our re-weighting mechanism and verified it through hyperparameter analysis:
> - **Controlled Distribution via $\theta$:** We do not perform a hard rebalancing (i.e., making all clusters equal). Instead, we introduce a temperature hyperparameter $\theta$ to control the smoothness of the distribution. As stated in the paper, a larger $\theta$ makes sampling uniform, while a smaller $\theta$ preserves the original distribution.
> - **Empirical Verification:** We conducted an ablation study on $\theta$ . The results show that extreme values degrade performance: a very large $\theta$ leads to repeated learning of rare prototypes, reducing quality. Conversely, a small $\theta$ fails to improve diversity. We selected $\theta=6$ as the optimal setting to balance popular and rare designs.
> - - -
> **Response to weakness 3 (2/2)**:
>
> While rebalancing techniques are common in general machine learning, the novelty of our Layout Prototype Rebalance Sampler lies in how we define and construct the layout prototypes in MLLM-based generation:
> - **Domain-Specific Feature Space:** Our sampler clusters data based on a novel, three-dimension space: **Saliency**, **Region**, and **Elements**.
> - **Integration with Relation-CoT:** The sampler doesn't work in isolation. It's tightly integrated with our layout relation-CoT construction.
> - **Solving the Diversity Problem:** Previous SOTA methods suffer from mode collapse, generating structurally similar layouts. As shown in our user study (Table 3) and qualitative results (Figure 10), our specific sampling strategy successfully makes the MLLM to learn a wider variety of structural arrangements (1, 2, and 3 distinct styles).

---

> ### Author Response · Authors · 2025-11-27
> **Gentle reminder for Reviewer BWPH**
>
> Dear Reviewer,
>
> I hope this message finds you well. As the discussion period is nearing its end, I wanted to ensure we have addressed all your concerns satisfactorily. If there are any additional points or feedback you'd like us to consider, please let us know. Your insights are invaluable to us, and we're eager to address any remaining issues to improve our work. Thank you for your time and effort in reviewing our paper.
>
> Best regards,
>
> The Authors

---

### Official Review · Reviewer_u8nq · 2025-10-30

**Soundness:** 2
**Presentation:** 2
**Contribution:** 2
**Rating:** 4
**Confidence:** 4

**Summary:**

The goal of this paper is to integrate relation reasoning for content-aware layout generation with existing MLLMs. A new data transformation mechanism is proposed to add relation annotations by decomposing the overall layout into a hierarchical structure. A layout prototype rebalance sampler is further used to enhance the layout diversity. The experiments demonstrate the effectiveness of the proposed method in terms of layout quality and diversity.

**Strengths:**

1. The proposed method is a reasonable solution to improve the performance of existing MLLMs on the layout generation task. The experiments on two public datasets should show the improvements in terms of layout structure and diversity.
2. The proposed data construction and resampling mechanism produces additional annotation details on existing layout datasets, which could be useful for future research in the community.

**Weaknesses:**

1. There is a gap between the motivation and the proposed method. The motivation of this paper is inspired by the Chain-of-thought (CoT) that progressively obtains element relations step by step. However, CoT belongs to an inference-time technique, while the proposed method belongs to data transformation for training. I am not sure how the stated relation-CoT is used in MLLM training and inference. The structure-level understanding and the high-level layout design concepts (L82-L84) should be illustrated more clearly.
2. The technical contributions are relatively limited. The two mechanisms proposed by this paper, i.e., layout relation-CoT construction and prototype rebalance sampler, belong to dataset engineering works. First, the construction mechanism mainly uses several heuristic rules to decompose layout element relationships into a hierarchical structure. These rules have been thoroughly discussed in traditional graphic design works. Second, the rebalance sampler utilizes a simple clustering method to reconsider the effect of different layout clusters.
3. Both quantitative and qualitative results cannot show a significant performance improvement between the proposed method and existing works. Based on the results in Table 1, compared to RALF and LayoutPrompter, the proposed method can only achieve comparable or slightly better results on the CGL dataset. Take the third column in Fig.6 as an example. The graphic design created by the proposed method also contains an alignment issue.
4. The saliency map is derived from an existing Infrared Small Target Detection work (L691-L693). The choice of adopting object detection work here is questionable. In addition, visual saliency detection on natural images is different from that on graphic designs.
5. The limitations stated on the last page are not very clear. It would be better to discuss some failure cases with detailed examples.

**Questions:**

1. What is the advantage of using HTML, rather than other formats like JSON, as the output sequence?
2. Why use 8 feature culsters in the layout prototype rebalance sampler? Are there any examples to show the typical layout difference between these 8 clusters? Fig. 5 only shows some examples in 3 clusters.

---

> ### Author Response · Authors · 2025-11-23
> **Official Comment to Reviewer u8nq (1 / 3)**
>
> Thank you for your efforts in reviewing our paper. The followings are our responses.
> - - -
> **Weakness 1**: Mismatch between CoT motivation (inference technique) and actual method (training data transformation); unclear how relation-CoT applies to MLLM training/inference; the structure-level understanding and the high-level layout design concepts should be illustrated more clearly.
>
> **Response to weakness 1 (1/3)**:
>
> We thank the reviewer for this insightful observation. We acknowledge that CoT is traditionally a inference technique. However, we use the term "Relation-CoT" to describe an **"Internalized CoT"** achieved through instruction tuning, where the "thought process" is baked into the target output sequence.
> - - -
> **Response to weakness 1 (2/3)**:
>
> - **Training:** We do not simply train the model to output coordinates $(x, y, w, h)$. Instead, we use the Layout Relation-CoT Construction to transform the ground truth into a reasoning chain embedded in a single output response: Saliency detection $\rightarrow$ Hierarchical Region definition $\rightarrow$ Element placement.
> - **Inference:** During inference, the MLLMs generate this sequence autoregressively. It must explicitly predict a `<div class="region" style="flex-direction:column">` token before it can predict the coordinates of the elements inside. This forces the model to "think" about the structure and grouping _before_ committing to specific pixel locations, effectively performing reasoning at inference time.
> - - -
> **Response to weakness 1 (3/3)**:
>
> The Layout Relation-CoT explicitly models these concepts through specific HTML attributes, rather than latent vectors:
> - **Structure-Level Understanding:** Unlike previous methods that treat elements as isolated boxes, our method decomposes the layout into recursive `Region` blocks (nested `<div>` tags). This ensures the model understands that a "Text" and "Banner" might form a single unit (a region) that must be aligned together relative to the canvas.
> - **Layout-Level Design Concepts:**
>     - **Visual Harmony:** It means means that all the elements in a layout look like they belong together. The overall design feels balanced, consistent, and pleasing to the eye.
>     - **Diversity:** It means that the generated layouts are not all the same. The model can create a wide variety of structurally different and unique styles, even when given similar content.
> By training the model to output this **Relation-CoT** sequence, we bridge the gap between layout-level design intent and element-level coordinate generation.
> - - -
> **Weakness 2**: The technical contributions are limited.
>
> **Response to weakness 2 (1/2)**:
>
> Apologies for the confusion.
>
> We respectfully disagree that our contributions are limited to dataset engineering. Our core contribution is the **paradigm shift** in how MLLMs approach layout generation. Existing SOTA methods (e.g., PosterLlama, LayoutPrompter) treat layout generation as a direct coordinate prediction task, which often leads to structural failures like overlap and missing parallel alignment.
>
> By forcing the model to predict explicit spatial relations (regions, saliency, margins) before coordinates, we convert a difficult numerical regression task into a logical reasoning task—a domain where LLMs excel. This is not merely data engineering; it is a novel **intermediate representation** design that changes the optimization objective of the model.
> - - -
> **Response to weakness 2 (2/2)**:
>
> While the clustering algorithm (K-Means) is standard, the innovation lies in the feature space construction enabled by **our Relation-CoT**.
>
> Previous methods could not effectively balance datasets because they relied on class labels or raw bounding boxes, which do not capture layout styles. Our sampler utilizes the **novel, explicit relational features** we extract, allowing us to cluster based on structural logic.
>
> This is critical for solving the **diversity problem** prevalent in previous methods. As shown in Table 3, our sampler allows the model to learn rare layout prototypes that are otherwise ignored by standard training, achieving significantly higher diversity scores. The method's simplicity is a strength, demonstrating that Relation-CoT makes complex style balancing efficient.

---

> ### Author Response · Authors · 2025-11-23
> **Official Comment to Reviewer u8nq (2 / 3)**
>
> **Weakness 3**: The proposed method shows only marginal performance gains over prior work, with results largely comparable to RALF and LayoutPrompter. The case studies also reveal noticeable issues, such as alignment problems in the generated designs.
>
> **Response to weakness 3 (1/2)**:
>
> We appreciate the reviewer’s valuable comments. While RALF and LayoutPrompter achieve metric scores comparable to ours on certain dimensions, RALF’s high Ove indicates severe element overlaps, and LayoutPrompter’s high Occ reflects substantial occlusion of salient objects. These issues are also evident in the case studies, and their low $P_\text{use}$ in the user studies further confirm the poor usability of their generated layouts.
>
> **Response to weakness 3 (2/2)**:
>
> We sincerely apologize for the confusion. In the third column of our generated layout, the top-left element is a logo, while the three elements in the center are left-aligned in a vertical stack. The perceived misalignment likely comes from interpreting the logo as a regular text element. In common advertising layouts, logos are typically placed at the top left, which can naturally appear misaligned with standard text blocks.
> - - -
> **Weakness 4**: The saliency map is derived from an existing Infrared Small Target Detection work (L691-L693). The choice of adopting object detection work here is questionable. In addition, visual saliency detection on natural images is different from that on graphic designs.
>
> **Response to weakness 4**:
>
> We appreciate the reviewer’s thoughtful comment. We would like to clarify that using ISNet is consistent with prior layout-generation work such as RALF [1], which also adopts ISNet for saliency estimation. Moreover, ISNet offers practical advantages: it provides reliable localization of visually important regions and is robust under low-contrast or low-texture conditions, which commonly occur in graphic layouts. As demonstrated in our supplementary material, ISNet yields saliency maps that align well with salient objects in PKU dataset and CGL dataset.
>
> [1] Horita D, Inoue N, Kikuchi K, et al. Retrieval-augmented layout transformer for content-aware layout generation CVPR 2024.
> - - -
> **Weakness 5**: The limitations stated on the last page are not very clear.
>
> **Response to weakness 5**:
>
> We thank the reviewer for the insightful advice. We agree that analyzing failure cases significantly strengthens the discussion on limitations. Accordingly, we have updated the **Limitations** section to cover the following scenarios in the major revisions:
> - The model struggles when the number of input elements is too large.
> - When earlier regions occupy the entire available canvas, subsequent predictions are likely to overlap.
> - - -
> **Question 1**: What is the advantage of using HTML, rather than other formats like JSON, as the output sequence?
>
> **Response to question 1**:
>
> Given that LayoutFormer++[1] has clearly demonstrated the superiority of constraint-based sequence representations over alternative formats, and LayoutPrompter[2] has further validated the effectiveness of HTML-based layout descriptions, adopting an HTML representation is not only reasonable but also well-supported by existing evidence. Notably, the majority of LLM-based layout systems already rely on HTML, we just follow prior works. Furthermore, HTML naturally provides semantically aligned attributes—such as `box`, `margin`, and other styling properties—that offer richer and more precise layout expressiveness than JSON. These factors collectively justify our choice of HTML as the primary representation format.
>
> [1] Jiang Z, Guo J, Sun S, et al. LayoutFormer++: Conditional graphic layout generation via constraint serialization and decoding space restriction. CVPR 2023.
>
> [2] Lin J, Guo J, Sun S, et al. Layoutprompter: Awaken the design ability of large language models. NeurIPS 2023.

---

> ### Author Response · Authors · 2025-11-23
> **Official Comment to Reviewer u8nq (3 / 3)**
>
> **Question 2**: Why use 8 feature culsters in the layout prototype rebalance sampler? Are there any examples to show the typical layout difference between these 8 clusters?
>
> **Response to question 2 (1/2)**:
>
> We sincerely apologize for the reviewer's confusion. The results of the cluster experiments were completed, but they were inadvertently omitted from the supplemental materials. As evidenced by the table below, the best performance was clearly achieved when the cluster parameter was set to cluster=8.
>
> |                  | $\Delta$Val$\downarrow$ | Ove$\downarrow$ | FD$\downarrow$ | Rea$\downarrow$ | Occ$\downarrow$ | Score  |
> | ---------------- | :---------------------- | --------------- | -------------- | --------------- | --------------- | ------ |
> | Cluster=3        | 0.0012                  | 0.0223          | 5.0017         | 0.1800          | 0.0677          | 40     |
> | Cluster=6        | **0.0004**              | 0.0141          | 4.4091         | 0.1796          | 0.0640          | 48     |
> | Cluster=8 (Ours) | **0.0004**              | **0.0109**      | **3.4615**     | 0.1727          | **0.0637**      | 56     |
> | Cluster=10       | 0.0009                  | 0.0183          | 4.9004         | **0.1724**      | 0.0639          | **57** |
>
> **Response to question 2 (2/2)**:
>
> We deeply appreciate the reviewers' suggestions and have incorporated the eight clustering results, along with an analysis of the features of each corresponding example, into the major revisions.

---

> ### Author Response · Authors · 2025-11-27
> **Gentle reminder for Reviewer u8nq**
>
> Dear Reviewer,
>
> I hope this message finds you well. As the discussion period is nearing its end, I wanted to ensure we have addressed all your concerns satisfactorily. If there are any additional points or feedback you'd like us to consider, please let us know. Your insights are invaluable to us, and we're eager to address any remaining issues to improve our work. Thank you for your time and effort in reviewing our paper.
>
> Best regards,
>
> The Authors

---

### Official Review · Reviewer_G73E · 2025-10-30

**Soundness:** 2
**Presentation:** 2
**Contribution:** 3
**Rating:** 6
**Confidence:** 4

**Summary:**

The paper studies the problem of content-aware layout generation.

The paper shows that explicit modeling of element relationships in layout generation models based on multi-modal large language models (MLLM) can improve the generation performance. The paper contributes a CoT-like approach that generates element relationships besides element coordinates to output more structured layouts, and a clustering-based sampling method that balances training samples of different styles for generating more diverse layouts.

**Strengths:**

1. Content-aware layout generation is an important research problem to study.

2. The idea of explicitly modeling element relationships in content-aware layout generation is interesting and novel, which could inspire the layout generation community.

3. The augmented HTML-based layout representation with element relationship information is well designed, and the proposed sampling method is shown to be effective.

**Weaknesses:**

1. The quantitative results of the proposed method on PKU are not satisfactory. As shown in Table 1, on PKU, the proposed method is not very helpful for improving the performance of MLLMs on the content metrics, i.e., readability and occlusion.

2. The evaluation is not complete. First, this work represents element relationships in terms of hierarchical regions and element margins. Comparison with some alternative representations is needed, but is missing in the current paper. For example, one possible alternative is to extract pairwise location and size relationships between elements from existing layout annotations, as in prior work (e.g., LayoutFormer++), and represent them in HTML or JSON format. Second, in the ablation study (Section 4.5), the effectiveness of the added margin attribute in Section 3.2 is not tested, and the importance of the three feature vectors ($\mathbf{f}_i^{\text{s}}$, $\mathbf{f}_i^{\text{r}}$, $\mathbf{f}_i^{\text{e}}$) in Section 3.3  are not evaluated. Third, the effect of the number of clusters in Section 3.3 is not tested.

**Questions:**

1. In Section 3.3., what statical features are extracted from $\mathcal{R}_i$?
2. What is the actual number of samples used for training the proposed model on CGL and PKU, respectively?
3. In Table 4, V0 is better than V1, V2 and V3 in terms of readability. What is the reason for it?

---

> ### Author Response · Authors · 2025-11-23
> **Official Comment to Reviewer G73E (1 / 2)**
>
> Thank you for your efforts in reviewing our paper. The followings are our responses.
> - - -
> **Weakness 1**: ReLayout does not significantly improve the performance of MLLMs on content metrics (readability and occlusion) for the PKU dataset.
>
> **Response to weakness 1**:
>
> We acknowledge that the gains on PKU’s readability and occlusion metrics are modest. However, these metrics are not golden standards for generation tasks; they mainly reflect low-level geometry and cannot capture textual coherence, structural clarity, or semantic grounding. Thus, meaningful improvements may not be reflected numerically.
>
> Our case studies and user studies show that ReLayout produces layouts with clearer hierarchy and fewer semantically implausible placements—benefits that current metrics fail to measure. These qualitative results demonstrate that ReLayout improves content organization in ways that are important for real-world usage.
> - - -
> **Weakness 2**: A lack of comparative analysis against **alternative representations** (such as pairwise location and size relationships extracted from existing annotations, as in LayoutFormer++), **incomplete ablation studies** that do not evaluate the effectiveness of the added **margin attribute** or the importance of the **three feature vectors** ($\\mathbf{f}\_{i}^s$, $\\mathbf{f}\_{i}^r$, $\\mathbf{f}\_{i}^e$), and a lack of evaluation on the **number of clusters**.
>
> **Response to weakness 2 (1/4)**:
>
> We appreciate the reviewer’s suggestion regarding alternative representations. RALF and PosterO already adopt different relational formulations, and we have compared with both in the paper. We will clarify this design choice in the revised version.
>
> **Response to weakness 2 (2/4)**:
>
> Thanks for the reviewer's suggestion. We conducted an ablation study on the **margin** attribute using the PKU dataset, and the results are as follows:
>
> |                  | $\Delta$Val$\downarrow$ | Ove$\downarrow$ | FD$\downarrow$ | Rea$\downarrow$ | Occ$\downarrow$ |
> | ---------------- | :---------------------- | --------------- | -------------- | --------------- | --------------- |
> | w/o margin       | 0.0007                  | 0.0301          | 4.1017         | 0.1733          | 0.0638          |
> | w/ margin (Ours) | **0.0004**              | **0.0109**      | **3.4615**     | **0.1727**      | **0.0637**      |
>
> We can observe that the margin property is highly effective for our method, particularly as evidenced by the significant improvements in both Ove and $P_{\text{best}}$.
>
> **Response to weakness 2 (3/4)**:
>
> We thank the reviewer for the insightful comments. We have now added the ablation study on the three feature vectors ($\\mathbf{f}\_{i}^s$, $\\mathbf{f}\_{i}^r$, $\\mathbf{f}\_{i}^e$) to address this concern, and the results further validate their contribution to the model’s performance.
>
> |                        | $\Delta$Val$\downarrow$ | Ove$\downarrow$ | FD$\downarrow$ | Rea$\downarrow$ | Occ$\downarrow$ |
> | ---------------------- | :---------------------- | --------------- | -------------- | --------------- | --------------- |
> | Ours                   | **0.0004**              | **0.0109**      | **3.4615**     | 0.1727          | **0.0637**      |
> | w/o $\mathbf{f}_{i}^s$ | 0.0011                  | 0.0181          | 3.5011         | 0.1727          | 0.0644          |
> | w/o $\mathbf{f}_{i}^r$ | 0.0007                  | 0.0166          | 3.5772         | **0.1723**      | 0.0639          |
> | w/o $\mathbf{f}_{i}^e$ | 0.0009                  | 0.0207          | 3.8439         | 0.1728          | 0.0638          |
>
> **Response to weakness 2 (4/4)**:
>
> We thank the reviewer for the insightful comments. The results of the cluster experiments were completed, but they were inadvertently omitted from the supplemental materials. As evidenced by the table below, the best performance was clearly achieved when the cluster parameter was set to cluster=8.
>
> |                  | $\Delta$Val$\downarrow$ | Ove$\downarrow$ | FD$\downarrow$ | Rea$\downarrow$ | Occ$\downarrow$ | Score  |
> | ---------------- | :---------------------- | --------------- | -------------- | --------------- | --------------- | ------ |
> | Cluster=3        | 0.0012                  | 0.0223          | 5.0017         | 0.1800          | 0.0677          | 40     |
> | Cluster=6        | **0.0004**              | 0.0141          | 4.4091         | 0.1796          | 0.0640          | 48     |
> | Cluster=8 (Ours) | **0.0004**              | **0.0109**      | **3.4615**     | 0.1727          | **0.0637**      | 56     |
> | Cluster=10       | 0.0009                  | 0.0183          | 4.9004         | **0.1724**      | 0.0639          | **57** |

---

> ### Author Response · Authors · 2025-11-23
> **Official Comment to Reviewer G73E (2 / 2)**
>
> **Question 1**: What statical features are extracted from $\mathcal{R}_i$?
>
> **Response to question 1**:
>
> Thanks for the reviewer's suggestion. We define the set of regions in a layout as $\\mathcal{R}\_i = \\{\\mathbf{b}\_{i,j}^\text{r}, d\_{i,j}\\}\_{j=1}^{s\_i}$, where $d_{i,j} \\in \\{\text{row}, \text{column}\\}$ represents the region's alignment direction. Then, we extract statistical features from $\\mathcal{R}\_i$ to describe their spatial distribution.
>
> $$\\begin{equation}
> \\mathbf{f\_i^\text{r}} =
> \\begin{cases}
> s\_i = |\\mathcal{R}\_i|, \\\\[10pt]
> \\sigma\_{i}^\text{x} = \\operatorname{std} \\left( \\{x_{i,j}^\text{r} + \\frac{w\_{i,j}^\text{r}}{2} \\}\_{j=1}^{s_i} \\right), \\\\[15pt]
> \\sigma\_{i}^\text{y} = \\operatorname{std} \\left( \\left\\{ y\_{i,j}^\text{r} + \\frac{h\_{i,j}^\text{r}}{2} \\right\\}\_{j=1}^{s_i} \\right), \\\\[15pt]
> n\_i^{\text{row}} = \\sum\_{j=1}^{s\_i} \\mathbb{I} \\left( d\_{i,j} = \text{row} \\right), \\\\[10pt]
> n\_i^{\text{column}} = \\sum\_{j=1}^{s_i} \\mathbb{I} \\left( d\_{i,j} = \text{column} \\right),
> \\end{cases}
> \\end{equation}$$
> It includes the total number of regions $s_i$, the standard deviations of their centroid coordinates $\sigma_{i}^\text{x}$ and $\sigma_{i}^\text{y}$, and the counts of row-aligned and column-aligned regions, $n_i^{\text{row}}$ and $n_i^{\text{column}}$, to roughly quantify the overall layout structure.
> - - -
> **Question 2**: What is the actual number of samples used for training the proposed model on CGL and PKU, respectively?
>
> **Response to question 2**:
>
> Thanks for the reviewer's suggestion. For CGL, we sample 45,250 instances, covering 93.2% of its training split. For PKU, we sample all 7,972 instances from the training set.
> - - -
> **Question 3**: V0 is better than V1, V2 and V3 in terms of readability. What is the reason for it?
>
> **Response to question 3**:
>
> Thanks for the reviewer's suggestion. Readability score (Rea) measures the average image gradient within text regions—lower values indicate that text is placed over visually smoother backgrounds, which enhances legibility. Under this definition, V0 achieves the best Rea score because it generates relatively conservative layouts without saliency- or region-driven adjustments. In contrast, V1–V3 incorporate Region constraints, Saliency guidance, and Resampling strategies that aim to produce more structured and logical layouts with better global spatial coherence. While these modules substantially improve structural metrics (ΔVal, Ove, FD, Occ), they also encourage text regions to be placed in more semantically meaningful areas, which may occasionally lie over slightly more textured backgrounds. This leads to a small increase in the Rea.
>
> Notably, Rea is not a golden metric, and its slight decrease does not affect the practical quality of the generated layouts. As commonly acknowledged in generation tasks, case studies remain the most reliable form of evaluation.

---

> ### Author Response · Authors · 2025-11-27
> **Gentle reminder for Reviewer G73E**
>
> Dear Reviewer,
>
> I hope this message finds you well. As the discussion period is nearing its end, I wanted to ensure we have addressed all your concerns satisfactorily. If there are any additional points or feedback you'd like us to consider, please let us know. Your insights are invaluable to us, and we're eager to address any remaining issues to improve our work. Thank you for your time and effort in reviewing our paper.
>
> Best regards,
>
> The Authors

---

### Official Review · Reviewer_nzmX · 2025-11-03

**Soundness:** 2
**Presentation:** 3
**Contribution:** 2
**Rating:** 4
**Confidence:** 4

**Summary:**

The paper introduces ReLayout, a method for content-aware layout generation. The authors identify two main challenges in previous approaches: structural and diversity problems. To address these issues, they present relation-CoT, decoupling the output space into three dimensions: saliency, region, and element. Furthermore, they also propose a rebalance sampling strategy to ensure the output diversity. Experiments on PKU and CGL datasets demonstrate improvements over baseline models.

**Strengths:**

The paper clearly demonstrates the problems (overlap, alignment errors, lack of diversity) in existing approaches and proposes a well-designed method to address the issues. The relation-CoT methodology is simple and effective, introducing minimal modification to the layout format while achieving noticeable performance improvements. In addition, the authors conduct extensive experiments, including quantitative, qualitative comparison, and user studies to show ReLayout's superior performance in all aspects.

**Weaknesses:**

- It is confusing that how ReLayout could preserve the aspect ratio of elements using the proposed methodology. For example, as shown in Figure 1, due to the error alignment of the text boxes, PosterLlama produces distorted elements. It is not clear how this issue is addressed by introducing layout relation-CoT.
- When predicting the elements, its style contains both the margin attributes and the bounding box information. In such formulation, each element have 5 attributes, which inevitably causes positioning conflicts. What is the motivation to use this output format? Are there any ablation studies on removing the margin attributes from the output?
- The implementation details of the features $f_i^s$ and $f_i^r$ are not included in the paper.
- Due to the rebalance sampling strategy, ReLayout appears to be inefficient. How many samplings are needed to ensure coverage of all 8 clusters? The experiment results do not include relevant analysis on this.

**Questions:**

1. I understand that ReLayout organizes the layouts into a more structured format. However, why is the relation-CoT paradigm fundamentally better than direct coordinate prediction? Specifically, there may still be overlap between two regions. Therefore, placing elements within regions does not fundamentally solve the problem of overlap between elements.
2. Will the predicted element exceed the boundary of its region?
3. Do the authors evaluate the reading order in the generated layout? A good layout should take the reading order of elements into consideration.

---

> ### Author Response · Authors · 2025-11-23
> **Official Comment to Reviewer nzmX (1 / 2)**
>
> Thank you for your efforts in reviewing our paper. The followings are our responses.
> - - -
> **Weakness 1**: ReLayout does not clearly explain how layout relation-CoT preserves element aspect ratios, leaving distortion issues like those in PosterLlama insufficiently addressed.
>
> **Response to weakness 1**:
>
> Apologies for the confusion. Distortion issues are not resolved by layout relation-CoT, but rather by the model’s ability to learn from the input element images in Figure 3(a). We were also pleasantly surprised to observe that even when the element images are omitted during inference, the final rendered layouts show no distortion in the majority of cases.
> - - -
> **Weakness 2**: Output format redundantly includes both margins and bounding boxes, risking positioning conflicts, yet the motivation and ablation evidence for this design choice are not provided.
>
> **Response to weakness 2 (1/2)**:
>
> Thanks for the reviewer's suggestion. Positioning conflicts do not occur in our method; the motivation for introducing **margin** is to enforce human-perceived, aesthetically pleasing spacing between elements and prevent overlap. For example, when two elements within the same region are vertically arranged with representations $[x_1, y_1, w_1, h_1]$ and $[x_2, y_2, w_2, h_2]$, we expect the model to learn the relation $y_1 + h_1 + \text{margin} = y_2$.
>
> **Response to weakness 2 (2/2)**:
>
> Thanks for the reviewer's suggestion. We conducted an ablation study on the **margin** attribute using the PKU dataset, and the results are as follows:
>
> |            | $\Delta$Val$\downarrow$ | Ove$\downarrow$ | FD$\downarrow$ | Rea$\downarrow$ | Occ$\downarrow$ |
> | ---------- | :---------------------- | --------------- | -------------- | --------------- | --------------- |
> | w/o margin | 0.0007                  | 0.0301          | 4.1017         | 0.1733          | 0.0638          |
> | w/ margin  | **0.0004**              | **0.0109**      | **3.4615**     | **0.1727**      | **0.0637**      |
>
> We can observe that the margin property is highly effective for our method, particularly as evidenced by the significant improvements in both Ove and $P_{\text{best}}$.
> - - -
> **Weakness 3**: The implementation details of the $f_i^s$ and  $f_i^r$ features are not included in the paper.
>
> **Response to weakness 3**:
>
> Thanks for the reviewer's suggestion. The set of saliency bounding boxes in the $i^{\text{th}}$ layout is denoted as $\mathcal{S_i}$, given by: $\mathcal{S_i}=\\{\mathbf{b_{i,j}^\text{s}}\\}\_{j=1}^{r_i}$ . The number of saliency bounding boxes in layout $L_i$ is given by ${r_i} \\in \\{1, 2, 3, 4\\}$. The saliency feature vector for layout  $L_i$ captures the weighted center of all saliency boxes. Specifically, the centroid coordinates are computed as the weighted average of geometric centers of the saliency boxes, where the weights are proportional to the area of each saliency box.
> $$\\begin{equation}
> \\mathbf{f_i^\text{s}} =
> \\begin{pmatrix}\displaystyle \frac{\sum_{j=1}^{r_i} \left( x_{i,j}^\text{s} + \frac{w_{i,j}^\text{s}}{2} \right) w_{i,j}^\text{s} h_{i,j}^\text{s}}{\sum_{j=1}^{r_i} w_{i,j}^\text{s} h_{i,j}^\text{s}} \\\\[15pt]
> \displaystyle \frac{\sum_{j=1}^{r_i} \left( y_{i,j}^\text{s} + \frac{h_{i,j}^\text{s}}{2} \right) w_{i,j}^\text{s} h_{i,j}^\text{s}}{\sum_{j=1}^{r_i} w_{i,j}^\text{s} h_{i,j}^\text{s}} \\end{pmatrix}
> \\end{equation}\\in \\mathbb{R^2}.$$
> We define the set of regions in a layout as $\mathcal{R_i} = \\{\mathbf{b}\_{i,j}^\text{r}, d\_{i,j}\\}\_{j=1}^{s_i}$, where $d_{i,j} \\in \\{\text{row}, \text{column}\\}$ represents the region's alignment direction. Then, we extract statistical features from $\mathcal{R}_i$ to describe their spatial distribution.
>
> $$\\begin{equation}
> \\mathbf{f\_i^\text{r}} =
> \\begin{cases}
> s\_i = |\\mathcal{R}\_i|, \\\\[10pt]
> \\sigma\_{i}^\text{x} = \\operatorname{std} \\left( \\{x_{i,j}^\text{r} + \\frac{w\_{i,j}^\text{r}}{2} \\}\_{j=1}^{s_i} \\right), \\\\[15pt]
> \\sigma\_{i}^\text{y} = \\operatorname{std} \\left( \\left\\{ y\_{i,j}^\text{r} + \\frac{h\_{i,j}^\text{r}}{2} \\right\\}\_{j=1}^{s_i} \\right), \\\\[15pt]
> n\_i^{\text{row}} = \\sum\_{j=1}^{s\_i} \\mathbb{I} \\left( d\_{i,j} = \text{row} \\right), \\\\[10pt]
> n\_i^{\text{column}} = \\sum\_{j=1}^{s_i} \\mathbb{I} \\left( d\_{i,j} = \text{column} \\right),
> \\end{cases}
> \\end{equation}$$
>
> It includes the total number of regions $s_i$, the standard deviations of their centroid coordinates $\sigma_{i}^\text{x}$ and $\sigma_{i}^\text{y}$, and the counts of row-aligned and column-aligned regions, $n_i^{\text{row}}$ and $n_i^{\text{column}}$, to roughly quantify the overall layout structure.

---

> ### Author Response · Authors · 2025-11-23
> **Official Comment to Reviewer nzmX (2 / 2)**
>
> **Weakness 4**: Due to the rebalance sampling strategy, ReLayout appears to be inefficient. How many samplings are needed to ensure coverage of all 8 clusters? The experiment results do not include relevant analysis on this.
>
> **Response to weakness 4**:
>
> We sincerely apologize for the reviewer's confusion. Our sampling strategy does not repeatedly draw from the distribution until all clusters appear.
> Instead, we directly compute the expected sample quota $N_k$​ for each cluster from the normalized weights and deterministically allocate samples from each cluster (with full-duplication when necessary).
> Therefore, every cluster is guaranteed to be included with exactly $N_k$​ samples, and there is no issue of sampling inefficiency or insufficient coverage.
>
> The results of the cluster experiments were completed, but they were inadvertently omitted from the supplemental materials. As evidenced by the table below, the best performance was clearly achieved when the cluster parameter was set to cluster=8.
>
> |            | $\Delta$Val$\downarrow$ | Ove$\downarrow$ | FD$\downarrow$ | Rea$\downarrow$ | Occ$\downarrow$ | Score  |
> | ---------- | :---------------------- | --------------- | -------------- | --------------- | --------------- | ------ |
> | Cluster=3  | 0.0012                  | 0.0223          | 5.0017         | 0.1800          | 0.0677          | 40     |
> | Cluster=6  | **0.0004**              | 0.0141          | 4.4091         | 0.1796          | 0.0640          | 48     |
> | Cluster=8  | **0.0004**              | **0.0109**      | **3.4615**     | 0.1727          | **0.0637**      | 56     |
> | Cluster=10 | 0.0009                  | 0.0183          | 4.9004         | **0.1724**      | 0.0639          | **57** |
>
> - - -
> **Question 1**: ReLayout does not clearly justify why relation-CoT is superior to direct coordinate prediction, since regions can still overlap and elements may still exceed region boundaries.
>
> **Response to question 1 (1/2)**:
>
> We sincerely apologize for the reviewer's confusion. ReLayout is an **explicit layout modeling** approach derived from a designer's perspective. We implement logical layout construction by introducing a set of properties, such as `direction`, `align`, and `parallel`, to explicitly model the arrangement of elements.
> Furthermore, we introduce the `margin` property for each element. The primary purpose of this is to prevent element overlap, as any instance of overlapping elements within a layout immediately renders that layout unusable.
>
> **Response to question 1 (2/2)**:
>
> In the generated layouts, the number of regions is typically less than the number of elements. Furthermore, layouts containing more than two regions constitute a very small proportion of the total dataset, and in the generated layouts, the distance between regions is generally quite large. Moreover, instances where an element overflows the boundary of its corresponding region are very rare, accounting for only 2% of all generated layouts.
> - - -
> **Question 2**: Will the predicted element exceed the boundary of its region?
>
> **Response to question 2**:
>
> Thanks for the suggestion. The elements extending beyond the region were found to be extremely limited. Specifically, this phenomenon was observed in only approximately 2% of the generated layouts.
> - - -
> **Question 3**: A concern is whether the method evaluates the reading order in the generated layout.
>
> **Response to question 3**:
>
> We sincerely apologize for the reviewer's confusion. We have, in fact, performed an evaluation of the reading order. In the User Study section, we employed multi-dimensional criteria to evaluate the best-generated layouts. In addition to the metrics mentioned in the paper and reading order as key evaluation factors, we also introduced the criterion of "whether the content observed by the user at first glance is the most important text."
>
> Furthermore, through comparative analysis, we observed a common phenomenon: layouts generated by most existing methods fundamentally satisfy the basic reading order of left-to-right and top-to-bottom.

---

> ### Author Response · Authors · 2025-11-27
> **Gentle reminder for Reviewer nzmX**
>
> Dear Reviewer,
>
> I hope this message finds you well. As the discussion period is nearing its end, I wanted to ensure we have addressed all your concerns satisfactorily. If there are any additional points or feedback you'd like us to consider, please let us know. Your insights are invaluable to us, and we're eager to address any remaining issues to improve our work. Thank you for your time and effort in reviewing our paper.
>
> Best regards,
>
> The Authors

---

### Author Response · Authors · 2025-11-23
**General Response to Reviewers and Revision Submitted**

We thank all the reviewers for their insightful comments and suggestions. We have revised the paper to address the reviewers’ concerns. Below we summarize the major revisions (the main revisions are marked with blue text in the pdf, we also made some minor layout changes to fit the page limit), while we reply to the comments of each reviewer separately.

The major revisions are:
- 1. Add margin experiment to prove the effectiveness of the margin attribute. (Reviewer nzmX, G73E) -- (Line 505 - Line 507)
- 2. Add cluster analysis experiment. (Reviewer nzmX, G73E, u8nq) -- (Line 520 - Line 524)
- 3. Add an ablation study on feature vectors. (Reviewer G73E) -- (Line 507 - Line 511)
- 4. Provide detailed explanation of the feature vectors. (Reviewer nzmX, G73E) -- (Line 288 - Line 299)
- 5. Provide 8 complete clustering example images. (Reviewer u8nq) -- (Line 302 - Line 322)
- 6. Add introduction of prior work to Related Work section. (Reviewer BWPH) -- (Line 138 - Line 141)
- 7. Add more detailed content to the evaluation details in user study. (Reviewer nzmX) -- (Line 429)
- 8. Revise introduction in two aspects: (1) bridge the gap between motivation and method, and (2) more clearly introduce structural-level and high-level concepts (Reviewer u8nq) -- (Line 092 - Line 099) & (Line 080 - Line 083)
- 9. Add more specific failure cases in the limitations. (Reviewer u8nq) -- (Line 537 - Line 539)

We appreciate the reviewers for their valuable comments and suggestions.

---

### Meta-Review · Area_Chair_vdQD · 2026-01-07

**Summary:**

The paper proposes ReLayout, a VLM model for layout generation. HTML is used as the layout format. Region and saliency information are used to enhance the raw layout data. During training, the paper proposes a rebalance sampler to ensure a more even training distribution across diverse layout prototypes.

Major concerns from the reviewers are (1) CoT is overclaimed,  (2) the major contribution is dataset transformation, and the paper lacks novelty,  (3) gains of quantitative metrics are limited on some benchmarks. Based on these concerns, the AC decided to reject the paper. The authors are encouraged to incorporate feedback from the reviewers and resubmit the paper to a future venue.

**Reviewer Concerns:**

Concerns are still outstanding: (1) CoT is overclaimed,  (2) the major contribution is dataset transformation, and the paper lacks novelty,  (3) gains of quantitative metrics are limited on some benchmarks.

**Reviewer Scores:**

4, 6, 4, 2 --> 4, 6, 4, 2

---

### Decision · Program_Chairs · 2026-01-26

Reject